# SPATIO-TEMPORAL GRAPH KNOWLEDGE DISTILLATION

## ABSTRACT

Large-scale spatio-temporal prediction is a critical area of research in data-driven urban computing, with far-reaching implications for transportation, public safety, and environmental monitoring. However, the challenges of scalability and generalization continue to pose significant obstacles. While many advanced models rely on Graph Neural Networks (GNNs) to encode spatial and temporal correlations, they often struggle with the increased time and space complexity of large-scale datasets. The recursive GNN-based message passing schemes used in these models can make their training and deployment difficult in real-life urban sensing scenarios. Additionally, large-scale spatio-temporal data spanning long time spans introduce distribution shifts, further highlighting the need for models with improved generalization performance. To address these challenges, we propose Spatio-Temporal Graph Knowledge Distillation (STGKD) paradigm to learn lightweight and robust Multi-Layer Perceptrons (MLPs) through effective knowledge distillation from cumbersome spatio-temporal GNNs. To ensure robust knowledge distillation, we integrate the spatio-temporal information bottleneck with the teacher-bounded regression loss. This allows us to filter out task-irrelevant noise and avoid erroneous guidance, resulting in robust knowledge transfer. Additionally, we enhance the generalization ability of student MLP by incorporating spatial and temporal prompts to inject downstream task contexts. We evaluate our framework on three large-scale spatio-temporal datasets for various urban computing tasks. Experimental results demonstrate that our model outperforms state-of-the-art approaches in terms of both efficiency and accuracy.

## 1 INTRODUCTION

Spatio-temporal prediction is the ability to analyze and model the complex relationships between spatial and temporal data. This involves understanding how different spatial features (*e.g.*, location, distance, and connectivity) and temporal features (*e.g.*, time of day, seasonality, and trends) interact with each other to produce dynamic patterns and trends over time. By accurately predicting these patterns and trends, spatio-temporal prediction enables a wide range of applications in urban computing. For example, in transportation, it can be used to predict traffic flow and congestion patterns, optimize traffic signal timing, and improve route planning for public transit systems Zheng et al. (2020b). In public safety, it can be used to predict crime hotspots and allocate police resources more effectively Xia et al. (2021). In environmental monitoring, it can be used to predict air and water quality, monitor the spread of pollutants, and predict the impact of climate change Yi et al. (2018).

Traditional spatio-temporal forecasting techniques often overlook the spatial dependencies present in data Yao et al. (2018; 2019); Pan et al. (2019); Shi et al. (2015). The emergence of Graph Neural Network (GNN)-based models Yu et al. (2018); Fang et al. (2021); Han et al. (2021); Wu et al. (2020) are motivated by the need to capture high-order spatial relationships between different locations, thereby enhancing the forecasting accuracy. By incorporating multiple graph convolutional or attention layers with recursively message passing frameworks, these models can effectively model the interactions among spatially connected nodes Geng et al. (2019). However, two key challenges hinder the performance of existing solutions in GNN-based spatio-temporal forecasting:

**Scalability**. Spatio-temporal prediction often involves large-scale datasets with complex spatial and temporal relationships. However, the computational complexity of GNNs can become prohibitive in

such cases. Specifically, GNN-based models for spatio-temporal prediction can be computationally demanding and memory-intensive due to the large-scale spatio-temporal graph they need to handle.

**Generalization**. Spatio-temporal prediction models need to generalize well to unseen data and adapt to distribution shifts that occur over time due to various factors, such as changes in the environment, human behavior, or other external factors Zhou et al. (2023). These distribution shifts can lead to a significant decrease in the performance of spatio-temporal prediction models Zhang et al. (2022). Therefore, it is important to consider spatio-temporal data distribution shift to ensure that the models can adapt to changes in the underlying distribution and maintain their accuracy over time.

**Contribution**. To tackle the aforementioned challenges, we propose our Spatio-Temporal Graph Knowledge Distillation paradigm (STGKD) that enables the transfer of knowledge from a larger, more complex teacher spatio-temporal GNN to a smaller, more efficient student model. This compression improves model scalability and efficiency, allowing for faster training and inference on resource-constrained systems in dealing with large-scale spatio-temporal data. Simultaneously, we focus on capturing and modeling the accurate and invariant temporal and spatial dependencies to enhance generalization capabilities. This enables the lightweight student model i) to be robust against noisy or irrelevant information after knowledge distillation from the teacher GNN; and ii) to adapt to distribution shifts when dealing with downstream unseen spatio-temporal data.

In the realm of spatio-temporal predictions, two types of noise can hinder the effectiveness of knowledge distillation: errors or inconsistencies in the teacher model's predictions and shifts in data distribution between training and testing data. Mitigating these biases in the teacher model's predictions and effectively handling data distribution shifts are crucial for achieving successful spatio-temporal knowledge distillation. This process holds the potential to improve the scalability and efficiency of spatio-temporal prediction models while enhancing their generalization capabilities. To accomplish this, we incorporate the principle of the spatio-temporal information bottleneck into the knowledge distillation framework, aiming to enhance model generalization and robustness. To prevent the student model from being misled by erroneous regression results from the teacher model, we employ a teacher-bounded regression loss for robust knowledge alignment.

Additionally, to further enhance the student model's performance on downstream tasks by incorporating spatio-temporal contextual information, we utilize spatio-temporal prompt learning. This approach allows us to provide explicit cues that guide the model in capturing spatial and temporal patterns in unseen data, effectively imparting task-specific knowledge to the compressed model. The evaluation results demonstrate the effectiveness of our proposed method, which has the potential to significantly improve efficiency and accuracy in various spatio-temporal prediction tasks in urban computing domains. For reproducibility purposes, we have made our model implementation available at the following anonymous link: https://anonymous.4open.science/r/STGKD.

## 2 PRELIMINARIES

**Spatio-Temporal Units**. Different urban downstream tasks may employ varying strategies for generating spatio-temporal units. For instance, in the domain of crime forecasting, the urban geographical space is often partitioned into $N = I \times J$ grids, where each grid represents a distinct region $r_{i,j}$. Spatio-temporal signals, such as crime counts, are then collected from each grid at previous $T$ time intervals. On the other hand, when modeling traffic data, spatio-temporal traffic volume signals are gathered using a network of sensors (*e.g.*, $r_i$), with data recorded at specific time intervals ($t \in T$).

**Spatio-Temporal Graph Forecasting**. The utilization of a Spatio-Temporal Graph (STG) $\mathcal{G}(\mathcal{V}, \mathcal{E}, \mathbf{A}, \mathbf{X})$ provides an effective means of capturing the relationships among different spatio-temporal units. In this context, $\mathcal{V}$ is the collection of nodes (*e.g.*, regions or sensors) and $\mathcal{E}$ denotes the set of edges that connect these nodes. The adjacency matrix, $\mathbf{A} \in \mathbb{R}^{N \times N}$ (where $N = |\mathcal{V}|$), captures the relationships between the nodes in the spatio-temporal graph. $\mathbf{X} \in \mathbb{R}^{T \times N \times F}$ represents the STG features, which encompass spatio-temporal signals such as traffic flow or crime counts. Here, $T$ signifies the number of time steps, while $F$ denotes the number of features associated with each node. This graph-based structure allows for an efficient characterization of spatial and temporal relationships, enabling a comprehensive analysis of the underlying urban dynamics. Our goal in STG prediction is to learn a function, denoted as $f$, that can forecast the future STG signals (*i.e.*, $\hat{\mathbf{Y}} \in \mathbb{R}^{T' \times N \times F}$) for the next $T'$ steps based on the available information from $T$ historical frames.

$$\hat{\mathbf{Y}}_{t:t+T'-1} = f(\mathcal{G}(\mathcal{V}, \mathcal{E}, \mathbf{A}, \mathbf{X}_{t-T:t-1})) \tag{1}$$

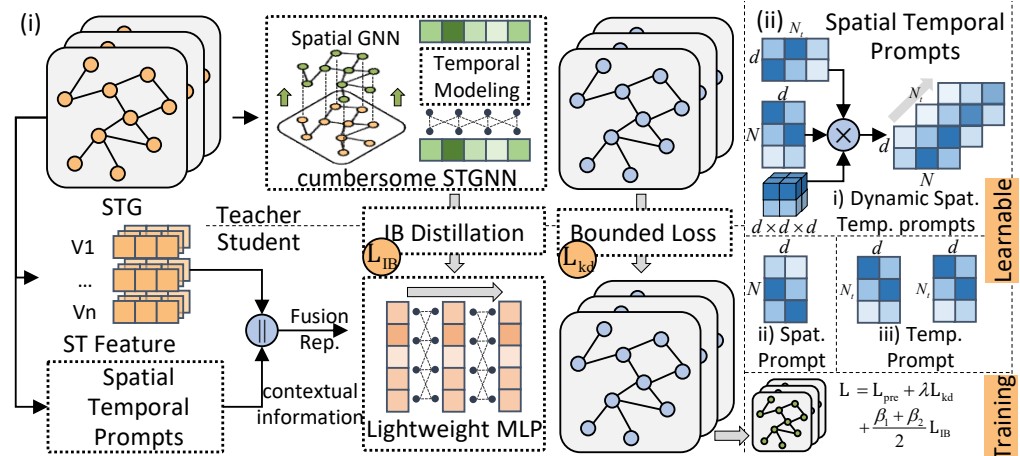

Figure 1: Overall framework of the proposed STGKD.

## 3 METHODOLOGY

In this section, we present our STGKD along with its technical details, as illustrated in Figure 1. Throughout this section, subscripts are used to represent matrix indices, while superscripts are employed to indicate specific distinguishing labels, unless stated otherwise.

### 3.1 KNOWLEDGE DISTILLATION WITH SPATIO-TEMPORAL GNNS

The effectiveness of spatio-temporal GNNs heavily relies on complex network models with recursive message passing schemes. In our STGKD, we aim to overcome this complexity by transferring the soft-label supervision from a large teacher model to a lightweight student model, while still preserving strong performance in spatio-temporal prediction. The teacher spatio-temporal GNN provides supervision through spatio-temporal signals (*i.e.,* $\mathbf{Y} \in \mathbb{R}^{T' \times N \times F}$), and it generates predictive labels (*i.e.,* $\mathbf{Y}^T \in \mathbb{R}^{T' \times N \times F}$). Our goal is to distill the valuable knowledge embedded in the GNN teacher and effectively transfer it to a simpler MLP, enabling more efficient and streamlined learning.

$$\mathcal{L} = \mathcal{L}_{\text{pre}}(\hat{\mathbf{Y}}, \mathbf{Y}) + \lambda \mathcal{L}_{\text{kd}}(\hat{\mathbf{Y}}, \mathbf{Y}^T) \tag{2}$$

The prediction of the student MLP is denoted as $\hat{\mathbf{Y}} \in \mathbb{R}^{T' \times N \times F}$. We introduce the trade-off coefficient $\lambda$ to balance the two terms in our objective. The first term, $\mathcal{L}_{\text{pre}}$, represents the predictive MAE-based or MSE-based loss function used in the original STG forecasting tasks. However, when it comes to knowledge distillation, the second term, $\mathcal{L}_{\text{kd}}$, which aims to bring the student's predictions closer to the teacher's results, requires careful reconsideration, especially for regression tasks. In the following subsection, we will present our well-designed objective that addresses this issue.

### 3.2 ROBUST KNOWLEDGE TRANSFER WITH INFORMATION BOTTLENECK

In the context of spatio-temporal predictions, the presence of two types of noise can indeed have a detrimental impact on the effectiveness of the knowledge distillation process. The predictions produced by the teacher model can be prone to errors or inconsistencies, which can misguide the knowledge transfer paradigm during the distillation process. Additionally, the presence of data distribution shift between the training and test data can pose a challenge for knowledge distillation. This can result in the student model struggling to identify relevant information for the downstream prediction task. As a result, addressing bias in the teacher model's predictions and handling data distribution shift are important considerations for successful spatio-temporal knowledge distillation.

To address the above challenges, we enhance our spatio-temporal knowledge distillation paradigm with Information Bottleneck principle (IB), to improve the model generalization and robustness. In particular, our objective of our framework in information compression is to generate compressed representations of input data that retains the invariant and most relevant information while discarding unnecessary or redundant information. Formally, we aim to minimize the objective by considering the student's predictions, denoted as $\hat{\mathbf{Y}}$, the teacher's predictions, denoted as $\mathbf{Y}^T$, the ground-truth

result, denoted as $\mathbf{Y}$, and the input spatio-temporal features, denoted as $\mathbf{X}$.

$$\min_{\mathbb{P}(\mathbf{Z}|\mathbf{X})} (-I(\mathbf{Y}, \mathbf{Z}) + \beta_1 I(\mathbf{X}, \mathbf{Z})) + (-I(\mathbf{Y}^T, \mathbf{Z}) + \beta_2 I(\mathbf{X}, \mathbf{Z}))$$

$$= \min_{\mathbb{P}(\mathbf{Z}|\mathbf{X})} -(I(\mathbf{Y}, \mathbf{Z}) + I(\mathbf{Y}^T, \mathbf{Z})) + (\beta_1 + \beta_2) I(\mathbf{X}, \mathbf{Z}) \tag{3}$$

The hidden representation, denoted as $\mathbf{Z}$, represents the encoded information of the input $\mathbf{X}$ in the student model. To incorporate certain constraints in the objective function, we introduce Lagrange multipliers $\beta_1$ and $\beta_2$. In our IB-enhanced knowledge distillation paradigm, we conduct two channels of distillation. The first channel aligns the predictions of the teacher model with those of the student model, while the second channel aligns the predictions of the student model with the downstream labels. By striking a balance between compression and relevance, our framework enables the discovery of compressed representations that capture the most salient and informative aspects of the data, while discarding irrelevant or redundant information.

### 3.2.1 VARIATIONAL BOUNDS OUR IB MECHANISM

Since directly computing the mutual information terms $I(\mathbf{Y}, \mathbf{Z})$, $I(\mathbf{Y}^T, \mathbf{Z})$, and $I(\mathbf{X}, \mathbf{Z})$ is intractable, we resort to using variational bounds to estimate each term in the objective, as motivated by the work Alemi et al. (2017). Concerning the lower bound of $I(\mathbf{Y}, \mathbf{Z}) + I(\mathbf{Y}^T, \mathbf{Z})$, its formalization can be expressed as follows:

$$I(\mathbf{Y}, \mathbf{Z}) + I(\mathbf{Y}^T, \mathbf{Z}) = \mathbb{E}_{\mathbf{Y}, \mathbf{Z}}[\log \frac{\mathbb{P}(\mathbf{Y}|\mathbf{Z})}{\mathbb{P}(\mathbf{Y})}] + \mathbb{E}_{\mathbf{Y}^T, \mathbf{Z}}[\log \frac{\mathbb{P}(\mathbf{Y}^T|\mathbf{Z})}{\mathbb{P}(\mathbf{Y}^T)}]$$

$$\geq \mathbb{E}_{\mathbf{Y}, \mathbf{Z}}[\log \mathbb{Q}_1(\mathbf{Y}|\mathbf{Z})] + \mathbb{E}_{\mathbf{Y}^T, \mathbf{Z}}[\log \mathbb{Q}_2(\mathbf{Y}^T|\mathbf{Z})] \tag{4}$$

The variational approximations $\mathbb{Q}_1(\mathbf{Y}|\mathbf{Z})$ and $\mathbb{Q}_2(\mathbf{Y}^T|\mathbf{Z})$ are used to approximate the true distributions $\mathbb{P}(\mathbf{Y}|\mathbf{Z})$ and $\mathbb{P}(\mathbf{Y}^T|\mathbf{Z})$, respectively. These approximations aim to closely match the ground-truth result $\mathbf{Y}$ and mimic the behavior of the teacher model $\mathbf{Y}^T$ based on the hidden embeddings $\mathbf{Z}$. As for the upper bound of $I(\mathbf{X}, \mathbf{Z})$, we can express it as follows:

$$I(\mathbf{X}, \mathbf{Z}) = \mathbb{E}_{\mathbf{X}, \mathbf{Z}}[\log \frac{\mathbb{P}(\mathbf{Z}|\mathbf{X})}{\mathbb{P}(\mathbf{Z})}] \leq \mathbb{E}_{\mathbf{X}}[\mathrm{KL}(\mathbb{P}(\mathbf{Z}|\mathbf{X})\|\mathbb{Q}_3(\mathbf{Z}))] \tag{5}$$

The variational approximation $\mathbb{Q}_3(\mathbf{Z})$ is used to approximate the marginal distribution $\mathbb{P}(\mathbf{Z})$. Further derivation and details can be found in the supplementary material. In our spatio-temporal IB paradigm, the objective to be minimized is given by Equation 3.

$$\min_{\mathbb{P}(\mathbf{Z}|\mathbf{X})} -(\mathbb{E}_{\mathbf{Y}, \mathbf{Z}}[\log \mathbb{Q}_1(\mathbf{Y}|\mathbf{Z})] + \mathbb{E}_{\mathbf{Y}^T, \mathbf{Z}}[\log \mathbb{Q}_2(\mathbf{Y}^T|\mathbf{Z})])$$

$$+ (\beta_1 + \beta_2)\mathbb{E}_{\mathbf{X}}[\mathrm{KL}(\mathbb{P}(\mathbf{Z}|\mathbf{X})\|\mathbb{Q}_3(\mathbf{Z}))] \tag{6}$$

### 3.2.2 SPATIO-TEMPORAL IB INSTANTIATING

To instantiate the objective in Eq 6, we characterize the following distributions: $\mathbb{P}(\mathbf{Z}|\mathbf{X})$, $\mathbb{Q}_1(\mathbf{Y}|\mathbf{Z})$, $\mathbb{Q}_2(\mathbf{Y}^T|\mathbf{Z})$, and $\mathbb{Q}_3(\mathbf{Z})$. These distributions play a crucial role in defining and instantiating the objective in Eq 6, allowing us to optimize the model based on the information bottleneck principle.

**Encoder with $\mathbb{P}(\mathbf{Z}|\mathbf{X})$.** To obtain the mean and variance matrices of the distribution of $\mathbf{Z}$ from the input feature $\mathbf{X}$, we employ a Multilayer Perceptron (MLP) encoder $\mathcal{F}_e$. The formulation is:

$$(\mu_z, \sigma_z) = \mathcal{F}_e(\mathbf{X}) \tag{7}$$

**Decoder with $\mathbb{Q}_1(\mathbf{Y}|\mathbf{Z})$ and $\mathbb{Q}_2(\mathbf{Y}^T|\mathbf{Z})$.** After obtaining the distribution of $\mathbf{Z}$ with mean $(\mu_z)$ and variance $(\sigma_z)$ matrices, we utilize the reparameterization trick to sample from this learned distribution and obtain the hidden representation $\mathbf{Z}$. The reparameterization is given by $\mathbf{Z} = \epsilon \sigma_z + \mu_z$, where $\epsilon$ is a stochastic noise sampled from a standard normal distribution ($\mathcal{N}(0, 1)$). Subsequently, we decode the obtained $\mathbf{Z}$ using an MLP decoder $\mathcal{F}_d$ to generate the final prediction $\hat{\mathbf{Y}}$:

$$\hat{\mathbf{Y}} = \mathcal{F}_d(\mathbf{Z}) \tag{8}$$

For tasks involving discrete predictions, such as classification, the cross-entropy loss is commonly used to maximize the likelihood in the first term of Equation 3. On the other hand, for regression tasks with continuous predictions, Equation 2 is employed, utilizing mean squared error (MSE) or mean absolute error (MAE) to maximize the likelihood. This choice of loss function depends on the nature of the prediction task and the type of output being considered.

**Marginal Distribution Control with $\mathbb{Q}_3(\mathbf{Z})$.** In our approach, we assume the prior marginal distribution of $\mathbf{Z}$ to be a standard Gaussian distribution $\mathcal{N}(0, 1)$. This choice is inspired by the spirit of variational auto-encoders (VAE) as discussed in the work Kingma & Welling (2014). Consequently, for the KL-divergence term in Equation 3, we can express it as follows:

$$\mathrm{KL}(\mathbb{P}(\mathbf{Z}|\mathbf{X})\|\mathbb{Q}_3(\mathbf{Z})) = \frac{1}{2}(-\log \sigma_z^2 + \sigma_z^2 + \mu_z^2 - 1) \tag{9}$$

The derivation of the above equation can be found in the supplementary materials.

### 3.2.3 Teacher-Bounded Regression Loss

To effectively control the knowledge distillation process for regression tasks, a teacher-bounded regression loss $\mathcal{L}_b$ is employed as the knowledge distillation loss $\mathcal{L}_{\mathrm{kd}}$. The purpose of this approach is to prevent the student model from being misled by deterministic yet erroneous regression results generated by the teacher model. The formulation of the teacher-bounded regression loss $\mathcal{L}_b$ is:

$$\mathcal{L}_{\mathrm{kd}}(\hat{\mathbf{Y}}, \mathbf{Y}^T) = \mathcal{L}_b(\hat{\mathbf{Y}}, \mathbf{Y}^T, \mathbf{Y}) = \begin{cases} \ell(\hat{\mathbf{Y}}, \mathbf{Y}), & \text{if } \ell(\hat{\mathbf{Y}}, \mathbf{Y}) + \delta \geq \ell(\mathbf{Y}^T, \mathbf{Y}) \\ 0, & \text{otherwise} \end{cases} \tag{10}$$

The symbol $\ell$ represents any standard regression loss, such as mean absolute error (MAE) or mean squared error (MSE). The threshold $\delta$ is used to control the knowledge transfer process. The vectors $\hat{\mathbf{Y}}$, $\mathbf{Y}^T$, and $\mathbf{Y}$ correspond to the predictions of the student, the teacher, and the ground truth, respectively. In detail, the student model does not directly take the teacher's predictions as its target but instead treats them as an upper bound. The objective of the student model is to approach the ground truth results and closely mimic the behavior of the teacher model. However, once the student model's performance surpasses that of the teacher model by a certain degree (exceeding the threshold $\delta$), it no longer incurs additional penalties for knowledge distillation. To conclude, we extend the original KD loss, which is constrained by the proposed spatio-temporal IB principle, resulting in a robust and generalizable KD framework. Our objective is to minimize the following function $\mathcal{L}$:

$$\mathcal{L} = \mathcal{L}_{\mathrm{pre}}(\hat{\mathbf{Y}}, \mathbf{Y}) + \lambda \mathcal{L}_{\mathrm{kd}}(\hat{\mathbf{Y}}, \mathbf{Y}^T) + \frac{\beta_1 + \beta_2}{2}(-\log \sigma_z^2 + \sigma_z^2 + \mu_z^2 - 1) \tag{11}$$

### 3.3 Spatio-Temporal Context Learning with Prompts

To infuse the spatio-temporal contextual information into the student model from downstream tasks, we leverage spatio-temporal prompt learning as a mechanism to impart task-specific knowledge to the compressed model. These prompts serve as explicit cues that guide the model in capturing data-specific spatial and temporal patterns. We incorporate the following spatio-temporal prompts:

**Spatial Prompt**. The diverse nodes present in the spatio-temporal graph showcase distinct global spatial characteristics, which are closely linked to the functional regions (*e.g.*, commercial and residential areas) they represent in urban geographical space. To effectively model this essential feature, we introduce a learnable spatial prompt denoted as $\mathbf{E}^{(\alpha)} \in \mathbb{R}^{N \times D}$, where $N$ denotes the number of nodes (*e.g.*, regions, sensors) within the spatio-temporal graph. This spatial prompt enables us to incorporate and encode the unique spatial characteristics associated with each spatial units.

**Temporal Prompt**. To further enhance the student's temporal awareness, we incorporate two temporal prompts into the model, taking inspiration from previous works Shao et al. (2022); Wu et al. (2019). These prompts include the "time of day" prompt, represented by $\mathbf{E}^{(ToD)} \in \mathbb{R}^{T_1 \times d}$, and the "day of week" prompt, represented by $\mathbf{E}^{(DoW)} \in \mathbb{R}^{T_2 \times d}$. The dimensionality of the "time of day" prompt is set to $T_1 = 288$, corresponding to 5-minute intervals, while the "day of week" prompt has a dimensionality of $T_2 = 7$ to represent the seven days of the week.

**Spatio-Temporal Transitional Prompt.** The spatial and temporal dependencies among nodes in the spatio-temporal graph can vary across different time periods, often reflecting daily mobility

patterns, such as peak traffic during morning and evening rush hours in residential areas due to commuting. Consequently, it becomes crucial to learn spatio-temporal context with transitional prompts for different timestamps. However, this task can be time-consuming and resource-intensive, particularly when dealing with large-scale datasets. Taking inspiration from the workHan et al. (2021), we tackle this challenge by scaling all timestamps to represent a single day. We then employ Tucker decomposition Tucker (1966) to learn the dynamic spatio-temporal transitional prompt for each node at all timestamps within a day, denoted as $N_t$.

$$\mathbf{E}_{t,n}^{(\beta)'} = \sum_{p=1}^{d} \sum_{q=1}^{d} \mathbf{E}_{p,q}^{k} \mathbf{E}_{t,p}^{t} \mathbf{E}_{n,q}^{s}, \qquad \mathbf{E}_{t,n}^{(\beta)} = \frac{\exp(\mathbf{E}_{t,n}^{(\beta)'})}{\sum_{m=1}^{N} \exp(\mathbf{E}_{t,m}^{(\beta)'})} \qquad (12)$$

Let $\mathbf{E}^{k} \in \mathbb{R}^{d \times d \times d}$ represent the Tucker core tensor with a Tucker dimension of $d$. We define $\mathbf{E}^{t} \in \mathbb{R}^{N_t \times d}$ to represent the temporal prompts, and $\mathbf{E}^{s} \in \mathbb{R}^{N \times d}$ to represent prompts for spatial locations. Additionally, $\mathbf{E}^{(\beta)'} \in \mathbb{R}^{N_t \times N \times d}$ and $\mathbf{E}^{(\beta)} \in \mathbb{R}^{N_t \times N \times d}$ indicate the intermediate and final prompts for spatio-temporal transitional patterns, respectively.

**Information Fusion with Spatio-Temporal Prompts and Representations.** To summarize, we aggregate spatio-temporal information from both prompts and latent representations to create the input $\mathbf{X}$ for the information bottleneck-regularize student model. The formal expression is as follows:

$$\mathbf{X} = \mathrm{FC}_1(\mathbf{X}) \| \mathrm{FC}_2(\mathbf{E}^{(\alpha)}) \| \mathrm{FC}_3(\mathbf{E}_{t-T,t-1}^{(\beta)}) \| \mathrm{FC}_4(\mathbf{E}_{t-T,t-1}^{(ToD)}) \| \mathrm{FC}_5(\mathbf{E}_{t-T,t-1}^{(DoW)}) \qquad (13)$$

Here, $\mathrm{FC}_i$, where $i = 1 \cdots 5$, refers to fully-connected layers that map all embeddings to the same dimensional space. The terms $\mathbf{E}_{t-T,t-1}^{(\beta)} \in \mathbb{R}^{T \times N \times d}$, $\mathbf{E}_{t-T,t-1}^{(ToD)} \in \mathbb{R}^{T \times d}$, and $\mathbf{E}_{t-T,t-1}^{(DoW)} \in \mathbb{R}^{T \times d}$ represent the learnable spatio-temporal prompts queried by the input "time of day" and "day of week" indices of the STG. After passing the student model according to Equations 7 and 8, we optimize our STGKD using Equation 11. For a more detailed explanation of the learning process of our STGKD framework, please refer to the Supplementary Materials.

## 4 EVALUATION

**Datasets**. To evaluate the effectiveness of our model in large-scale spatio-temporal prediction, we employ urban sensing datasets for three distinct tasks: traffic flow prediction, crime forecasting and weather prediction. i) **Traffic Data**. PEMS is a traffic dataset collected from the California Performance of Transportation (PeMS) project. It consists of data from 1481 sensors, with a time interval of 5 minutes. The dataset spans from Sep 1, 2022, to Feb 28, 2023. ii) **Crime Data**. CHI-Crime is a crime dataset obtained from crime reporting platforms in Chicago. For this dataset, we divide the city of Chicago into spatial units of size 1 km × 1 km, resulting in a total of 1470 grids. The time interval for this dataset is 1 day, covering the period from Jan 1, 2002, to Dec 31, 2022. ii) **Weather Data**. This is a weather dataset released by Zhu et al. (2023). It comprises data from 1866 sensors, with a temporal resolution of 1 hour. The dataset spans from Jan 1, 2017, to Aug 31, 2021. To show the superiority of our STGKD more intuitively, we also evaluate it on the public dataset PEMS-4. For more detailed statistics of those datasets, please refer to the Supplementary Section.

**Evaluation Protocols**. To ensure a fair comparison, we divided the three datasets into a ratio of 6:2:2 for training, validation, and testing, respectively. For traffic prediction, we specifically focused on the flow variable to perform our predictions. For crime forecasting, we select four specific crime types for our analysis. In the task of weather prediction, our attention was directed towards the vertical visibility variable. To evaluate the performance of our model on these datasets, we utilized three commonly adopted evaluation metrics: *Mean Absolute Error (MAE)*, *Root Mean Squared Error (RMSE)*, and *Mean Absolute Percentage Error (MAPE)*.

**Baseline Models.** We conducted a comparative analysis of our model against 12 state-of-the-art baselines. The baseline models include: (1) Statistical Approach: **HI** Cui et al. (2021); (2) Conventional Deep Learning Models: **MLP**, **FC-LSTM** Sutskever et al. (2014); (3) GNN-based Methods: **STGCN** Yu et al. (2018), **GWN** Wu et al. (2019), **StemGNN** Cao et al. (2021), **MTGNN** Wu et al. (2020); (4) Dynamic Graph-based Model: **DMSTGCN** Han et al. (2021); (5) Attention-based Method: **ASTGCN** Guo et al. (2019); (6) Hybrid Learning Model: **ST-Norm** Deng et al. (2021), **STID** Shao et al. (2022); (7) Self-Supervised Learning Approach: **ST-SSL** Ji et al. (2023).

Table 1: Performance comparison in diverse spatio-temporal forecasting tasks.

| Dataset | | Traffic | | | PEMS-04 | | | Crime | | | Weather | | |
|---|---|---|---|---|---|---|---|---|---|---|---|---|---|
| Model | Venue | MAE | RMSE | MAPE | MAE | RMSE | MAPE | MAE | RMSE | MAPE | MAE | RMSE | MAPE |
| HI | - | 34.62 | 55.51 | 26.38% | 42.35 | 61.66 | 29.92% | 1.0001 | 1.2221 | 82.84% | 6683.05 | 9532.07 | 114.35% |
| MLP | - | 19.16 | 33.80 | 13.69% | 26.34 | 40.53 | 17.53% | 0.8070 | 1.0098 | 64.92% | 4628.18 | 6854.31 | 78.34% |
| FC-LSTM | NeurIPS-14 | 18.22 | 32.75 | 13.43% | 23.81 | 36.62 | 18.12% | 0.8588 | 1.0541 | 69.72% | 4549.03 | 6895.66 | 77.99% |
| ASTGCN | AAAI-19 | 19.69 | 34.47 | 15.65% | 22.93 | 35.22 | 16.56% | 0.6584 | 0.9143 | 50.84% | 5891.46 | 8037.68 | 110.61% |
| STGCN | IJCAI-18 | 15.36 | 28.77 | 12.37% | 19.63 | 31.32 | 13.32% | 0.5749 | 0.8601 | 44.24% | 3997.19 | 6199.53 | 65.25% |
| GWN | IJCAI-19 | 14.10 | 27.14 | 9.80% | 19.22 | 30.74 | 12.52% | 0.6860 | 0.9165 | 55.88% | 3991.24 | 6207.5 | 65.63% |
| StemGNN | NeurIPS-20 | 13.97 | 27.26 | 9.73% | 21.61 | 33.80 | 16.10% | 0.7906 | 1.0095 | 63.69% | 4094.09 | 6370.02 | 68.43% |
| MTGNN | KDD-20 | 13.53 | 25.73 | 9.90% | 19.50 | 32.00 | 14.04% | 0.6551 | 0.9030 | 51.85% | 3991.14 | 6199.61 | 65.42% |
| ST-Norm | KDD-21 | 13.14 | 25.80 | 9.52% | 18.96 | 30.98 | 12.69% | 0.7727 | 1.0264 | 61.79% | 3996.73 | 6282.06 | 66.43% |
| DMSTGCN | KDD-21 | 14.50 | 27.86 | 9.97% | 22.87 | 36.05 | 14.86% | 0.7609 | 0.9778 | 60.92% | 4257.63 | 6554.1 | 71.15% |
| STID | CIKM-22 | 12.87 | 25.64 | 9.86% | 18.91 | 30.57 | 12.67% | 0.2337 | 0.6969 | 11.79% | 3997.92 | 6199.77 | 65.34% |
| ST-SSL | AAAI-23 | 14.49 | 26.48 | 12.38% | 20.88 | 32.69 | 13.95% | 0.3038 | 0.7045 | 18.59% | 3991.26 | 6250.69 | 67.90% |
| **STGKD** | **-** | **12.70** | **25.32** | **9.46%** | **18.69** | **30.46** | **12.34%** | **0.2281** | **0.6933** | **10.78%** | **3990.07** | **6195.83** | **65.08%** |

**Implementation Details.** The batch size for handling spatio-temporal data is set to 32. For model training, we initialize the learning rate at 0.002 and apply a decay factor of 0.5 with decay steps occurring at epochs 1, 50, and 100. Regarding the model's hyperparameters, $\beta_1$, $\beta_2$, $\lambda$ are chosen from (0.0, 1.0) to appropriately balance the various loss components. We designate the hidden dimension $d$ as 64, while the threshold $\delta$ for the bounded loss is determined as 0.1. In terms of the input-output sequence lengths for spatio-temporal prediction, we utilize the following configurations: i) *Traffic forecasting*: 12 historical time steps (1 hour) and 12 prediction time steps (1 hour). ii) *Crime prediction*: 30 historical time steps (1 month) and 1 prediction time step (1 day). ii) *Weather prediction*: 12 historical time steps (12 hours) and 12 prediction time steps (12 hours). We evaluate most of the baselines using their publicly available code with the default hyperparameter settings to ensure fair comparisons. The default teacher model employed in our experiments is STGCN.

## 4.1 PERFORMANCE COMPARISON

Table 1 presents the comparison results of our STGKD with state-of-the-art baselines on traffic, crime and weather information, evaluating its effectiveness. The best-performing model's results are highlighted in bold for each dataset. Overall, our STGKD has consistently demonstrated superior performance compared to various baselines, validating the effectiveness of our approach in modeling spatio-temporal correlations. The design of our IB-based spatio-temporal knowledge distillation paradigm enables the student MLP to inherit rich spatio-temporal knowledge from the teacher STGNN while avoiding erroneous guidance and potential noise from the teacher. This greatly enhances the model's ability to identify useful spatio-temporal correlations and outperforms GNN-based methods. Moreover, our framework achieves better generalization and robustness performance compared to self-supervised approaches (*e.g.*, ST-SSL), particularly on sparse crime data.

## 4.2 MODEL ABLATION STUDY

To verify the effectiveness of the designed modules, we perform comprehensive ablation experiments on key components of our model. The experimental results on three datasets are presented in Table 2. Accordingly, we have the following variants and observations:

- **Spatio-Temporal Prompt Learning**. We conduct experiments to remove the spatial, temporal and transitional prompts and generate three variants: "*w/o*-S-Pro", "*w/o*-T-Pro", "*w/o*-Tran-Pro", respectively. The results of these experiments show that all three types of prompts improve the model performance by injecting informative spatio-temporal contexts from the downstream tasks.

- **Spatio-Temporal IB**. We exclude the spatio-temporal IB module to create a model variant: "*w/o*-IB". Upon comparing the results across the three datasets, we note that the presence of our IB module enables the student model to extract and filter significant information in assisting the downstream spatio-temporal predictions, thereby improving generalization during the encoding and knowledge distillation. This effect is particularly pronounced in the sparse crime data.

- **Teacher-Bounded Regression Loss**. We substitute the bounded loss with the regular KD loss, specifically using the MAE loss ($\mathcal{L}_{kd}(\hat{\mathbf{Y}}, \mathbf{Y}^T)$), to create a model variant called "*w/o*-TB". Upon evaluation, we have observed a notable decrease in the performance of our STGKD. This outcome suggests that our designed teacher-bounded loss for alignment can effectively alleviate to transfer erroneous information from the teacher model to the student model.

- **Spatio-Temporal Knowledge Distillation**. To assess the effectiveness of our KD paradigm, we generate a model variant called "*w/o*-KD" by removing the knowledge distillation component. Upon evaluation, we have observed a significant decrease in the model's performance. This obser-

Table 2: Ablation study on various spatio-temporal forecasting tasks.

| Datasets | Traffic | | | Crime | | | Weather | | |
|---|---|---|---|---|---|---|---|---|---|
| Metrics | MAE | RMSE | MAPE | MAE | RMSE | MAPE | MAE | RMSE | MAPE |
| **STGKD** | **12.70** | **25.32** | **9.46%** | **0.2281** | **0.6933** | **10.78%** | **3990.07** | **6195.83** | **65.08%** |
| *w/o*-Tran-Pro | 12.85 | 25.41 | 9.95% | 0.2351 | 0.6977 | 10.80% | 4019.39 | 6196.10 | 65.18% |
| *w/o*-S-Pro | 13.28 | 25.73 | 9.56% | 0.2298 | 0.7779 | 10.80% | 4134.47 | 6330.35 | 67.35% |
| *w/o*-T-Pro | 13.81 | 26.08 | 10.52% | 0.2343 | 0.6952 | 11.40% | 4056.44 | 6200.58 | 65.85% |
| *w/o*-IB | 12.84 | 25.35 | 9.47% | 0.2733 | 0.7189 | 15.73% | 4006.01 | 6196.78 | 65.29% |
| *w/o*-TB | 13.14 | 25.68 | 10.25% | 0.2400 | 0.7000 | 11.42% | 4085.29 | 6198.84 | 68.11% |
| MLP | 19.16 | 33.80 | 13.69% | 0.8070 | 1.0098 | 64.92% | 4628.18 | 6854.31 | 78.34% |

vation further solidifies the effectiveness of our proposed framework, highlighting the importance of the spatio-temporal knowledge transfer process in improving the model's performance.

## 4.3 MODEL SCALABILITY STUDY

In order to evaluate the effectiveness and efficiency of our STGKD in addressing large-scale spatio-temporal prediction, we conduct a comparative analysis with state-of-the-art baselines on the forecasting tasks of citywide traffic flow and crimes. The performance and inference time on the test sets of these datasets are presented in Figure 2. From our analysis, we highlight two key observations: **(i) Higher Efficiency:** Our STGKD achieves significantly faster inference speeds compared to existing SOTA models. This efficiency is attributed to the absence of complex computational units with GNN-based message passing in the lightweight student MLP model, allowing for faster computations without compromising performance. **(ii) Superior Prediction Accuracy:** The student MLP selectively inherits task-relevant spatio-temporal knowledge from the teacher GNN framework through knowledge distillation with

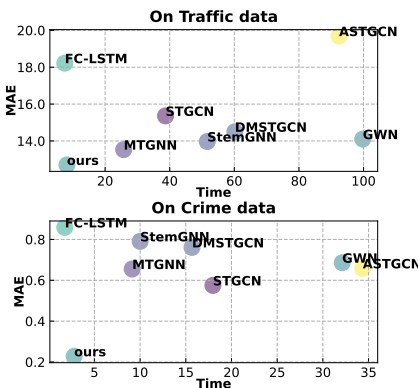

Figure 2: Model performance and inference time of representative methods on the test set of traffic and crime datasets.

our spatio-temporal IB paradigm and the teacher-bounded loss. These observations underscore the effectiveness and efficiency of our STGKD in large-scale spatio-temporal prediction tasks.

## 4.4 GENERALIZATION AND ROBUSTNESS STUDY

To further validate the robustness and generalization ability of our model, we compare it with baselines under the conditions of noisy and missing data over the PEMS traffic data. **Performance $w.r.t$ Data Noise:** We artificially introduce noise to the input STG features $\mathbf{X}$ by modifying the features as $\mathbf{X} = (1-\gamma)\mathbf{X}+\gamma\epsilon$, where $\gamma$ is the noise coefficient, and $\epsilon$ is sampled from a Gaussian distribution. We gradually increase the noise coefficient from 0 (original input) to 0.3 (with an increment of 0.05) and compare our model with STGCN, DMST-GCN, and MLP. The results, shown in Figure 3 (top), demonstrate that as the noise coefficient increases, the performance gap between DMSTGCN, MLP, and our model widens. Within the 0-0.2 range, the performance gap between STGCN and our model also continues to increase. This reflects the strong noise resilience of our model, where our spatio-temporal IB paradigm filters out task-irrelevant information. **Performance $w.r.t$**

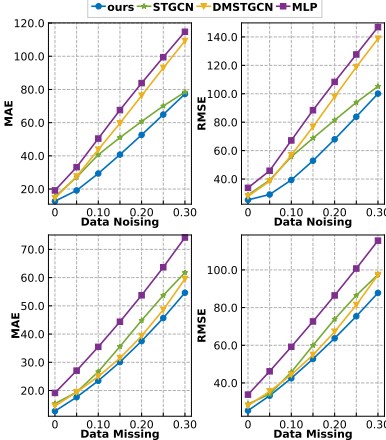

Figure 3: Performance evaluation $w.r.t$ noisy (top) and missing (bottom) data.

**Data Missing:** We manually set a certain proportion of the input STG features $\mathbf{X}$ to zero, simulating the data missing problem in real-world scenarios. The missing ratio is denoted as $\gamma = \frac{M}{T \times N \times F}$, where $M$ represents the total number of features in $\mathbf{X}$ that are set to zero. By gradually increasing the missing ratio from 0 (original input) to 0.3, Figure 3 (bottom) illustrates that the performance gap between the three comparison models and our model continues to widen. This further verifies

the superior ability of our model to learn robust and generalizable representations of STGs using limited features. Additionally, since our model does not require inter-feature message passing like STGNN, the impact of missing features on our model is minimized.

### 4.5 Model-agnostic Property Study

Our STGKD framework is model-agnostic, allowing it to be applied to different teachers. To validate its adaptability, we apply it to four STGNN models: STGCN, MTGNN, DMST-GCN, and StemGNN. The results on the traffic dataset are presented in Table 3. It can be observed that with the support of our framework, the performance of all teacher models is significantly improved, reaching the state-of-the-art level. This improvement can be attributed to our spatio-temporal IB and teacher-bounded loss, which effectively transfer task-relevant spatio-temporal knowledge to the student while filtering out noisy and misleading guidance. As a result, the positive effects of STGNN are maximized within our graph KD framework.

Table 3: Performance with various teacher models.

| Dataset | Traffic | | | Dataset | Traffic | | |
|---|---|---|---|---|---|---|---|
| Model | MAE | RMSE | MAPE | Model | MAE | RMSE | MAPE |
| STGCN | 15.36 | 28.77 | 12.37% | MTGNN | 13.53 | 25.73 | 9.90% |
| *w/*-KD | **12.70** | **25.32** | **9.46%** | *w/*-KD | **12.71** | **25.27** | **9.81%** |
| DMSTGCN | 14.50 | 27.86 | 9.97% | StemGNN | 13.97 | 27.26 | 9.73% |
| *w/*-KD | **12.76** | **25.23** | **9.57%** | *w/*-KD | **12.86** | **25.51** | **10.01%** |

## 5 Related Work

**Spatio-Temporal Forecasting**. In recent years, there have been significant advancements in spatio-temporal prediction within the domain of urban intelligence. This field enables accurate forecasting of complex phenomena such as traffic flow, air quality, and urban outliers. Researchers have developed a range of neural network techniques, including convolutional neural networks (CNNs) Zhang et al. (2017b;a), as well as graph neural networks (GNNs) Guo et al. (2019); Zheng et al. (2020a); Han et al. (2021). Moreover, recent self-supervised spatio-temporal learning methods (*e.g.*, ST-SSL Ji et al. (2023) and AutoST Zhang et al. (2023)) have shown great promise in capturing complex spatio-temporal patterns, especially in scenarios with sparse data. However, SOTA approaches still face challenges in terms of scalability and computational complexity when dealing with large-scale spatio-temporal graphs. Additionally, it is crucial for spatio-temporal prediction models to adapt well to distribution shifts over time in order to maintain their accuracy. This work aims to address these challenges by developing efficient and robust spatio-temporal forecasting frameworks.

**Knowledge Distillation on General Graphs**. Research on knowledge distillation (KD) for graph-based models has gained significant attention in recent years Zhang et al. (2021). The proposed paradigms of knowledge distillation can be grouped into two categories: i) *Logits Distillation* involves using logits as indicators of the inputs for the final softmax function, which represent the predicted probabilities. In the context of graph-based KD models, the primary objective is to minimize the difference between the probability distributions or scores of a teacher model and a student model. Noteworthy works that leverage logits in knowledge distillation for graphs include TinyGNN Yan et al. (2020), CPF Yang et al. (2021), and GFKD Deng & Zhang (2021). ii) *Structures Distillation* aims to preserve and distill either local structure information (*e.g.*, LSP Yang et al. (2020), FreeKD Feng et al. (2022), GNN-SD Chen et al. (2021)) or global structure information (*e.g.*, CKD Wang et al. (2022), GKD Yang et al. (2022)) from a teacher model to a student model. Notable examples in this category include T2-GNN Huo et al. (2022), SAIL Yu et al. (2022), and GraphAKD He et al. (2022). Drawing upon prior research, this study capitalizes on the benefits of KD to improve spatio-temporal prediction tasks. The objective is to streamline the process by employing a lightweight yet effective model. A significant contribution of this work lies in the novel integration of the spatio-temporal information bottleneck into the KD framework. By doing so, the model effectively mitigates the impact of noise through debiased knowledge transfer.

## 6 Conclusion

In our research, we focus on addressing two crucial challenges in large-scale spatio-temporal prediction: efficiency and generalization. To overcome these challenges, we introduce a novel and versatile framework called STGKD, which aims to encode robust and generalizable representations of spatio-temporal graphs. Our framework incorporates the IB principle to enhance the knowledge distillation process by filtering out task-irrelevant noise in the student's encoding and alignment during knowledge transfer. Moreover, we introduce a spatio-temporal prompt learning component that injects dynamic context from the downstream prediction task. Through extensive experiments, we demonstrate that our STGKD surpasses state-of-the-art models in both performance and efficiency.

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
