# A  SUPPLEMENTARY MATERIALS

In this supplementary material, i) We provide a detailed derivation of the proposed spatio-temporal IB principle in Section A.1. ii) We also delve into a comprehensive discussion of the reasons behind the robustness of their STGKD and provide insights into the computational complexity of their STGKD in Section A.2. iii) In Section A.3, we conduct an in-depth analysis of the learning process of the model in Section A.3, shedding light on its inner workings. iv) We showcase the detailed preprocessing steps and specific characteristics of the three large-scale datasets used in our work in Section A.4. v) In Section A.5, we elucidate the detailed information of the baselines used in the experiments. vi) In Section A.6, we present supplementary experimental results, including experimental questions, visualization of predictions, hyperparameter investigation, and a model interpretation case study. Overall, this supplementary material provides comprehensive insights and additional details to complement the main paper and facilitate a deeper understanding of our model.

## A.1  DERIVATIONS OF THE SPATIO-TEMPORAL INFORMATION BOTTLENECK PRINCIPLE

### A.1.1  DERIVATION FOR EQUATION 4

For the lower bound of $I(\mathbf{Y}, \mathbf{Z}) + I(\mathbf{Y}^T, \mathbf{Z})$, we have

$$
\begin{aligned}
I(\mathbf{Y}, \mathbf{Z}) + I(\mathbf{Y}^T, \mathbf{Z}) &= \int_{\mathbf{Z} \sim z, \mathbf{Y} \sim y} dy dz p(z,y) \log \frac{p(z,y)}{p(y)p(z)} + \int_{\mathbf{Z} \sim z, \mathbf{Y}^T \sim y^T} dy^T dz p(z,y^T) \log \frac{p(z,y^T)}{p(y^T)p(z)} \\
&= \int_{\mathbf{Z} \sim z, \mathbf{Y} \sim y} dy dz p(z,y) \log \frac{p(y|z)}{p(y)} + \int_{\mathbf{Z} \sim z, \mathbf{Y}^T \sim y^T} dy^T dz p(z,y^T) \log \frac{p(y^T|z)}{p(y^T)} \\
&= \mathbb{E}_{\mathbf{Y}, \mathbf{Z}}[\log \frac{\mathbb{P}(\mathbf{Y}|\mathbf{Z})}{\mathbb{P}(\mathbf{Y})}] + \mathbb{E}_{\mathbf{Y}^T, \mathbf{Z}}[\log \frac{\mathbb{P}(\mathbf{Y}^T|\mathbf{Z})}{\mathbb{P}(\mathbf{Y}^T)}]
\end{aligned} \tag{14}
$$

As we always have

$$
\text{KL}[\mathbb{P}(\mathbf{Y}|\mathbf{Z})\|\mathbb{Q}_1(\mathbf{Y}|\mathbf{Z})], \ \text{KL}[\mathbb{P}(\mathbf{Y}^T|\mathbf{Z})\|\mathbb{Q}_2(\mathbf{Y}^T|\mathbf{Z})] \geq 0
$$

$$
\int_{\mathbf{Y} \sim y} dy p(y|z) \log \frac{p(y|z)}{q_1(y|z)}, \int_{\mathbf{Y}^T \sim y^T} dy^T p(y^T|z) \log \frac{p(y^T|z)}{q_2(y^T|z)} \geq 0
$$

$$
\int_{\mathbf{Y} \sim y} dy p(y|z) \log p(y|z) \geq \int_{\mathbf{Y} \sim y} dy p(y|z) \log q_1(y|z),
$$

$$
\int_{\mathbf{Y}^T \sim y^T} dy^T p(y^T|z) \log p(y^T|z) \geq \int_{\mathbf{Y}^T \sim y^T} dy^T p(y^T|z) \log q_2(y^T|z)
$$

$$
\tag{15}
$$

where $\mathbb{Q}_1(\mathbf{Y}|\mathbf{Z})$ and $\mathbb{Q}_2(\mathbf{Y}^T|\mathbf{Z})$ indicates the arbitrary distribution of $\mathbf{Y}$ and $\mathbf{Y}^T$ given $\mathbf{Z}$. Substitute Equation 15 into Equation 14, then we obtain

$$
\begin{aligned}
I(\mathbf{Y}, \mathbf{Z}) + I(\mathbf{Y}^T, \mathbf{Z}) &\geq \int_{\mathbf{Z} \sim z, \mathbf{Y} \sim y} dy dz p(z,y) \log \frac{q_1(y|z)}{p(y)} + \int_{\mathbf{Z} \sim z, \mathbf{Y}^T \sim y^T} dy^T dz p(z,y^T) \log \frac{q_2(y^T|z)}{p(y^T)} \\
&= \int_{\mathbf{Z} \sim z, \mathbf{Y} \sim y} dy dz p(z,y) \log q_1(y|z) - \int_{\mathbf{Z} \sim z, \mathbf{Y} \sim y} dy dz p(z,y) \log p(y) + \\
&\quad \int_{\mathbf{Z} \sim z, \mathbf{Y}^T \sim y^T} dy^T dz p(z,y^T) \log q_2(y^T|z) - \int_{\mathbf{Z} \sim z, \mathbf{Y}^T \sim y^T} dy^T dz p(z,y^T) \log p(y^T) \\
&= \int_{\mathbf{Z} \sim z, \mathbf{Y} \sim y} dy dz p(z,y) \log q_1(y|z) - \int_{\mathbf{Y} \sim y} dy \log p(y) \int_{\mathbf{Z} \sim z} dz p(z,y) + \\
&\quad \int_{\mathbf{Z} \sim z, \mathbf{Y}^T \sim y^T} dy^T dz p(z,y^T) \log q_2(y^T|z) - \int_{\mathbf{Y}^T \sim y^T} dy^T \log p(y^T) \int_{\mathbf{Z} \sim z} dz p(z,y^T) \\
&= \int_{\mathbf{Z} \sim z, \mathbf{Y} \sim y} dy dz p(z,y) \log q_1(y|z) - \int_{\mathbf{Y} \sim y} dy p(y) \log p(y) + \\
&\quad \int_{\mathbf{Z} \sim z, \mathbf{Y}^T \sim y^T} dy^T dz p(z,y^T) \log q_2(y^T|z) - \int_{\mathbf{Y}^T \sim y^T} dy^T p(y^T) \log p(y^T) \\
&= \mathbb{E}_{\mathbf{Y}, \mathbf{Z}}[\log \mathbb{Q}_1(\mathbf{Y}|\mathbf{Z})] + H(\mathbf{Y}) + \mathbb{E}_{\mathbf{Y}^T, \mathbf{Z}}[\log \mathbb{Q}_2(\mathbf{Y}^T|\mathbf{Z})] + H(\mathbf{Y}^T)
\end{aligned} \tag{16}
$$

Because $H(\mathbf{Y})$ and $H(\mathbf{Y}^T)$ are constant given $\mathbf{Y}$ and $\mathbf{Y}^T$, we eventually obtain Equation 4.

### A.1.2 DERIVATION FOR EQUATION 5

Regarding the upper bound of $I(\mathbf{X}, \mathbf{Z})$, the formalization is as below:

$$
\begin{aligned}
I(\mathbf{X}, \mathbf{Z}) &= \int_{\mathbf{Z} \sim z, \mathbf{X} \sim x} dx dz p(z, x) \log \frac{p(x, z)}{p(x)p(z)} \\
&= \int_{\mathbf{Z} \sim z, \mathbf{X} \sim x} dx dz p(z, x) \log \frac{p(z|x)}{p(z)} \\
&= \mathbb{E}_{\mathbf{X}, \mathbf{Z}}[\log \frac{\mathbb{P}(\mathbf{Z}|\mathbf{X})}{\mathbb{P}(\mathbf{Z})}]
\end{aligned}
\tag{17}
$$

Since we always have

$$
\begin{aligned}
\mathrm{KL}[\mathbb{P}(\mathbf{Z})\|\mathbb{Q}_3(\mathbf{Z})] &\geq 0 \\
\int_{\mathbf{Z} \sim z} dz p(z) \log \frac{p(z)}{q_3(z)} &\geq 0 \\
\int_{\mathbf{Z} \sim z} dz p(z) \log p(z) &\geq \int_{\mathbf{Z} \sim z} dz p(z) \log q_3(z)
\end{aligned}
\tag{18}
$$

Substitute Eq 18 into Eq 17, and we have

$$
\begin{aligned}
I(\mathbf{X}, \mathbf{Z}) &= \int_{\mathbf{Z} \sim z, \mathbf{X} \sim x} dx dz p(z, x) \log p(z|x) - \int_{\mathbf{Z} \sim z, \mathbf{X} \sim x} dx dz p(z, x) \log p(z) \\
&= \int_{\mathbf{Z} \sim z, \mathbf{X} \sim x} dx dz p(z, x) \log p(z|x) - \int_{\mathbf{Z} \sim z} dz p(z) \log p(z) \\
&\leq \int_{\mathbf{Z} \sim z, \mathbf{X} \sim x} dx dz p(z, x) \log p(z|x) - \int_{\mathbf{Z} \sim z} dz p(z) \log q_3(z) \\
&= \int_{\mathbf{Z} \sim z, \mathbf{X} \sim x} dx dz p(z, x) \log p(z|x) - \int_{\mathbf{Z} \sim z, \mathbf{X} \sim x} dx dz p(x, z) \log q_3(z) \\
&= \int_{\mathbf{Z} \sim z, \mathbf{X} \sim x} dx dz p(z|x) p(x) \log \frac{p(z|x)}{q_3(z)} \\
&= \mathbb{E}_{\mathbf{X}}[\mathrm{KL}(\mathbb{P}(\mathbf{Z}|\mathbf{X})\|\mathbb{Q}_3(\mathbf{Z}))]
\end{aligned}
\tag{19}
$$

### A.1.3 DERIVATION FOR EQUATION 9

Suppose the prior marginal distribution, denoted as $\mathbb{Q}_3(\mathbf{Z})$, is assumed to follow a standard Gaussian distribution $\mathcal{N}(0, 1)$. Additionally, consider the encoder distribution $\mathbb{P}(\mathbf{Z}|\mathbf{X})$, which is modeled as a Gaussian distribution with mean and variance matrices represented by $\mu_z$ and $\sigma_z$ respectively.

$$
\begin{aligned}
\mathrm{KL}(\mathbb{P}(\mathbf{Z}|\mathbf{X})\|\mathbb{Q}_3(\mathbf{Z})) &= \int dx \frac{1}{\sqrt{2\pi\sigma_z^2}} \exp(-\frac{(x - \mu_z)^2}{2\sigma_z^2}) \log \frac{\frac{1}{\sqrt{2\pi\sigma_z^2}} \exp(-\frac{(x - \mu_z)^2}{2\sigma_z^2})}{\frac{1}{\sqrt{2\pi}} \exp(-\frac{x^2}{2})} \\
&= \int dx \frac{1}{\sqrt{2\pi\sigma_z^2}} \exp(-\frac{(x - \mu_z)^2}{2\sigma_z^2}) \log \frac{1}{\sqrt{\sigma_z^2}} \exp(\frac{x^2}{2} - \frac{(x - \mu_z)^2}{2\sigma_z^2}) \\
&= \int dx \frac{1}{\sqrt{2\pi\sigma_z^2}} \exp(-\frac{(x - \mu_z)^2}{2\sigma_z^2})[-\frac{1}{2} \log \sigma_z^2 + \frac{1}{2}x^2 - \frac{(x - \mu_z)^2}{2\sigma_z^2}] \\
&= \frac{1}{2}[-\log \sigma_z^2 + \mathbb{E}[x^2] - \frac{1}{\sigma_z^2}\mathbb{E}[(x - \mu_z^2)]] \\
&= \frac{1}{2}(-\log \sigma_z^2 + \sigma_z^2 + \mu_z^2 - 1)
\end{aligned}
\tag{20}
$$

## A.2 In-depth Discussion of our Proposed STGKD Framework

### A.2.1 Rationale Analysis of STGKD'Robustness

Previous methods for knowledge distillation (KD) on vanilla graphs have mainly focused on robustness in handling noise. For example, NOSMOG Tian et al. uses adversarial training to ensure that the student model is resilient to feature noise during KD. Similarly, GCRD Joshi et al. (2021) uses self-supervised contrastive learning to enhance robustness. However, our model takes a unique approach by prioritizing information control to achieve robust KD. The information control process within our KD framework plays a crucial role in determining the inherent robustness of KD.

In our proposed spatio-temporal IB principle, our STGKD aims to achieve simultaneous alignment of the encoded hidden representations $\mathbf{Z}$ with both the ground-truth $\mathbf{Y}$ and the teacher's predictions $\mathbf{Y}^T$ while reducing their correlation with the input spatio-temporal graph (STG) features $\mathbf{X}$. We posit that the input STG features are prone to noise originating from various sources, such as sensor malfunctions and inherent spatio-temporal distribution shifts. By mitigating the correlation between the hidden representations and the input features, our STGKD effectively captures environment-invariant information during the student encoding and teacher distillation process, thereby facilitating robust learning Alemi et al. (2017). To ensure tractability of the objective, we employ variational estimation and instantiate the spatio-temporal IB using a student MLP encoder and decoder.

During the training stage, we optimize the loss function expressed in Equation 11. TThe first term of the loss function aims to minimize the discrepancy between the predicted outputs of the student model and the ground-truth labels. The second term of the loss function minimizes the difference between the student's predictions and the teacher's predictions, promoting knowledge transfer from the teacher model to the student. The third term aims to reduce the correlation with the input spatio-temporal features. By jointly optimizing these terms, our model achieves robust KD by aligning the hidden representations with both the desired outputs and the teacher's knowledge while reducing their dependence on noisy input features.

### A.2.2 Model Complexity Analysis

In this analysis, we compare the time complexity of our STGKD with other state-of-the-art baselines. In many advanced STGNN models, Graph Convolutional Networks (GCNs) and self-attention mechanisms are commonly used to capture spatial correlations. Let's consider an L-layer GCN with fixed hidden features of size $d^{(s)}$. In STGNNs that utilize a predefined adjacency matrix, the time complexity is approximately $O^{(s)}(L \times |\mathcal{E}| \times d^{(s)} + L \times |\mathcal{V}| \times d^{(s)2})$. However, when an adaptive adjacency matrix is employed to enhance performance, the complexity becomes approximately $O^{(s)}(L \times |\mathcal{V}|^2 \times d^{(s)} + L \times |\mathcal{V}| \times d^{(s)2})$. Regarding the self-attention mechanism, models typically require $O^{(s)}(T \times |\mathcal{V}| \times d^{(s)})$ time complexity to compute the query, key, and value matrices. On the other hand, previous approaches often incorporate Temporal Convolutional Networks (TCNs) and self-attention to capture temporal dependencies. For an L-layer TCN with hidden feature dimension $d^{(t)}$, STGNNs require approximately $O^{(t)}(T \times |\mathcal{V}| \times d^{(t)} \times L)$ time complexity. In the case of self-attention, the time complexity for calculating the query, key, and value matrices is approximately $O^{(t)}(T \times |\mathcal{V}| \times d^{(t)})$. In contrast, our STGKD captures spatial and temporal correlations using a unified encoder-decoder MLP with a hidden dimension of $d$, input dimension of $d^{(in)}$, and output dimension of $d^{(out)}$. Therefore, the overall time complexity of our unified model is approximately $O(|\mathcal{V}| \times (d^{(in)} + d^{(out)}) \times d)$. Theoretically speaking, our STGKD exhibits significant computational complexity advantages compared to advanced STGNNs, thanks to its lightweight MLP architecture.

## A.3 Learning Process of the STGKD

We present the detailed learning process of our STGKD in Algorithm 1.

## A.4 Details of the Experimental Datasets

In this subsection, we present a detailed data processing procedure, followed by an overview of the data characteristics and methods for constructing graphs.

---

**Algorithm 1:** Learning Process of STGKD Framework

---

**Input:** spatio-temporal graphs (STG) $\mathcal{G}(\mathcal{V}, \mathcal{E}, \mathbf{A}, \mathbf{X})$, the trained teacher model $f_T$, regularization weight $\lambda$, Lagrange multipliers $\beta_1$ and $\beta_2$, maximum epoch number $E$, learning rate $\eta$

**Output:** trained parameters of the student in $\Theta$

1  Initialize all parameters of the student in $\Theta$
2  **for** $e = 1$ *to* $E$ **do**
    // Obtain the output of the teacher
3      $\mathbf{Y}^T = f_T(\mathcal{G}(\mathcal{V}, \mathcal{E}, \mathbf{A}, \mathbf{X}))$
    // The student training
4      Generate indexed spatio-temporal prompts $\mathbf{E}^{(\alpha)}$ and $\mathbf{E}^{(\beta)}_{\mathbf{t-T,t-1}}$, the *time of day* and *day of week* prompts $\mathbf{E}^{(ToD)}_{t-T,t-1}$ and $\mathbf{E}^{(DoW)}_{t-T,t-1}$.
5      Obtain the fused feature embeddings $\mathbf{X}$ based on Equation 13.
    // MLP encoder
6      Gain the mean and variance matrices $\mu_z$ and $\sigma_z$ of the distribution of $\mathbf{Z}$ from the input feature $\mathbf{X}$ according to Equation 7.
7      Sample a instantiated hidden representation $\mathbf{Z}$ using the reparameterization trick.
    // MLP decoder
8      Obtain the predictive results $\hat{\mathbf{Y}}$ of the student according to Equation 8.
9      Calculate the teacher bounded loss $\mathcal{L}_{kd}$ with $\hat{\mathbf{Y}}$ and $\mathbf{Y}^T$ based on Equation 10.
10     Calculate the Daul-Path IB loss $\mathcal{L}_{IB} = \frac{\beta_1+\beta_2}{2}(-\log \sigma_z^2 + \sigma_z^2 + \mu_z^2 - 1)$.
11     Calculate predictive loss $\mathcal{L}_{pre}$ with $\hat{\mathbf{Y}}$ and $\mathbf{Y}$.
12     Combine the loss terms together to get $\mathcal{L}$.
13     **for** *each parameter* $\theta \in \Theta$ **do**
14         $\theta = \theta - \eta \cdot \partial\mathcal{L}/\partial\theta$
15     **end**
16 **end**
17 **return** all parameters $\Theta$

---

### A.4.1 DATA PROCESSING

**Traffic Data**: We have collected traffic data from the 7th district of the California Performance of Transportation (PeMS) website Chen et al. (2001), specifically from September 1, 2022, to February 28, 2023, with a time interval of 5 minutes. The dataset comprises three variables: traffic flow, occupancy, and speed. To ensure the reliability of our experiments, we meticulously handpicked 1481 sensors that possess complete data without any missing values. In our analysis, we will focus on utilizing the traffic flow variable as both the observed and predictive factor.

**Crime Data**: We collected crime records from the official Chicago government platform [1], spanning over a period of 21 years, from January 1, 2002, to December 31, 2022. To analyze the data, we divided the city of Chicago into a grid of $42 \times 35$ spatial units, each measuring $1km \times 1km$. The dataset consists of four types of crimes: 'THEFT', 'BATTERY', 'ASSAULT', and 'CRIMINAL DAMAGE'. Our experiments included utilizing all four types of crime records as both observed and testing variables, to obtain a comprehensive understanding of the data.

**Weather Data**: We use the publicly available weather-2k dataset Zhu et al. (2023), which comprises data from 1866 weather stations, covering the period from January 1, 2017, to August 31, 2021. For our analysis, we have chosen vertical visibility as the observed and testing variable.

**PEMS-04**: The PEMS-04 dataset, cited as a benchmark dataset for spatio-temporal prediction in the STSGCN paper Song et al. (2020), was collected by the California Performance of Transportation (PeMS) system. This dataset comprises 307 spatial sensors and provides a time interval of 5 minutes. It covers the period from January 2018 to February 2018, offering valuable data for analyzing and predicting spatio-temporal patterns in transportation.

---

[1] https://data.cityofchicago.org/

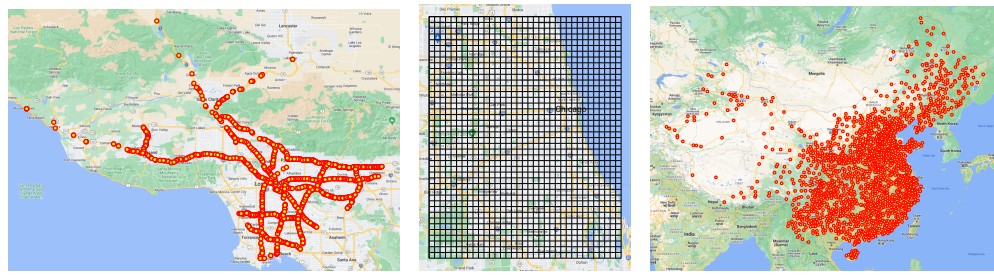

(a) Sensor distribution of PEMS traffic data     (b) Grid partitioning of crime data     (c) Sensor distribution of weather data

Figure 4: Visualization of the spatial distribution of different datasets.

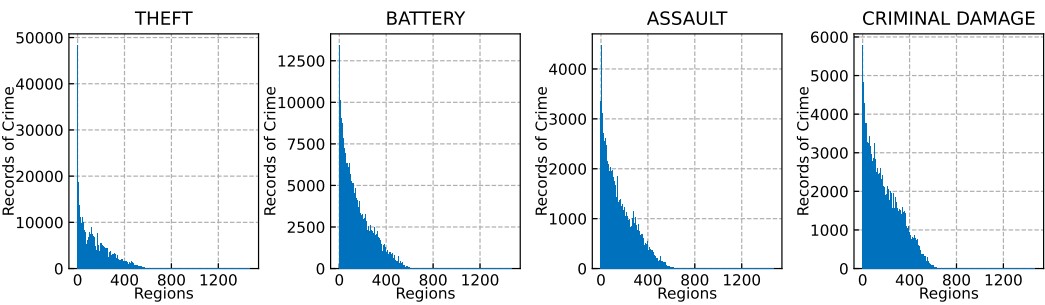

Figure 5: Distribution of crime occurrence at geographical regions on the crime dataset.

### A.4.2 CHARACTERISTICS OF EXPERIMENTAL DATASETS

The distribution of sensors in the PEMS traffic data and Weather data is visualized in Figure 5. In order to construct the spatial graph, we adopt a thresholded Gaussian kernel approach, inspired by the work of Li et al. (2018). The resulting adjacency matrix, denoted as $\mathbf{A}$, is defined as follows:

$$\mathbf{A}_{ij} = \exp(-\frac{\text{dist}(v_i, v_j)}{\sigma^2}), \text{ if dist}(v_i, v_j) \leq \kappa, \text{otherwise } 0 \tag{21}$$

The distance between the $i$-th station and the $j$-th station, denoted as $\text{dist}(v_i, v_j)$, is calculated to determine the spatial relationship. The parameters $\sigma$ and $\kappa$ are the standard deviation of distances and the threshold, respectively. In the case of PEMS traffic data, we set $\sigma = 100$ and $\kappa = 1e - 5$ during implementation. For Weather data, different values are used, specifically $\sigma = 15$ and $\kappa = 1e-1$. In the context of grid-based crime prediction, the construction process described in Section 2.

In Figure 4, we present a visualization of the total sum of various crime records across different regions. The distribution clearly demonstrates that the crime records in the dataset are skewed, with many regions having sparse crime records. This observation indicates that certain regions experience relatively higher crime rates compared to others, while the majority of regions have lower crime incidence. We further summarize the statistical information of there datasets in Table 4.

### A.5 DETAILS OF BASELINES

This subsection provides detailed explanations and implementations of the baselines used in our experiments. These baselines serve as comparison methods for evaluating the model performance.

Table 4: Statistical information of different datasets.

| Dataset | # regions | # time steps | # features | time span | interval | mean | min | max | media |
|---|---|---|---|---|---|---|---|---|---|
| Traffic | 1481 | 52128 | 1 | 2022/09/01 - 2023/02/28 | 5 min | 231.0731 | 0.0 | 1538.0 | 139.0 |
| Crime | 1470 | 7670 | 4 | 2002/01/01 - 2022/12/31 | 1 day | 0.0910 | 0.0 | 110.0 | 0.0 |
| Weather | 1866 | 40896 | 1 | 2017/01/01 - 2021/08/31 | 1 hour | 15906.7880 | 0.0 | 30000.0 | 14700.0 |

### A.5.1 DESCRIPTIONS OF BASELINES

The detailed descriptions of 11 baselines are as following:

**HI** Cui et al. (2021): This baseline uses the most recent time steps from the input as the direct predictive output for temporal modeling.

**MLP**: This approach employs a stacked multilayer perceptron (MLP) to capture the spatio-temporal correlations of the spatio-temporal sequences.

**FC-LSTM** Sutskever et al. (2014): Fully-Connected LSTM incorporates fully connected hidden units, enhancing its capacity to model and learn from complex temporal and sequential patterns.

**ASTGCN** Guo et al. (2019): This approach integrates the spatio-temporal attention mechanism, enabling it to capture dynamic spatio-temporal characteristics simultaneously.

**STGCN** Yu et al. (2018): It merges spatial graph convolutional networks and temporal gated convolutional layers to generate spatio-temporal representations for graph-structured time series.

**GWN** Wu et al. (2019): Graph WaveNet combines gated temporal convolutional layers and graph convolutional network layers to collectively capture both spatial and temporal dependencies.

**StemGNN** Cao et al. (2021): Spectral Temporal Graph Neural Network leverages inter-series correlations and temporal dependencies by jointly modeling them in the spectral domain.

**MTGNN** Wu et al. (2020): This approach enhances Graph WaveNet with mix-hop propagation for spatial modeling, dilated inception for temporal modeling, and refined graph learning layers for improved performance and capabilities in traffic prediction.

**ST-Norm** Deng et al. (2021): ST-Norm presents a novel factorization approach for multivariate time series data and proposes temporal normalization and spatial normalization techniques to enhance the high-frequency and local components of the MTS data.

**DMSTGCN** Han et al. (2021): The DMSTGCN method improves the graph convolutional network by incorporating dynamic spatial and temporal information. It utilizes a time-aware graph constructor to capture periodic patterns and dependencies among road segments.

**STID** Shao et al. (2022): This work proposes a simple yet effective baseline for spatio-temporal forecasting by attaching spatio-temporal identity information.

**ST-SSL** Ji et al. (2023): The ST-SSL framework includes a spatial self-supervised learning paradigm with adaptive graph augmentation and a clustering-based generative task. It also incorporates a temporal self-supervised learning paradigm with a time-aware contrastive task to capture spatio-temporal traffic patterns.

### A.5.2 BASELINE IMPLEMENTATIONS

The publicly available source codes for the baselines can be found at the following URLs: **HI**, **MLP**, **FC-LSTM**: https://github.com/zezhishao/BasicTS

**ASTGCN**: https://github.com/guoshnBJTU/ASTGCN-r-pytorch

**STGCN**: https://github.com/VeritasYin/STGCN_IJCAI-18

**GWN**: https://github.com/nnzhan/Graph-WaveNet

**StemGNN**: https://github.com/microsoft/StemGNN

**MTGNN**: https://github.com/nnzhan/MTGNN

**ST-Norm**: https://github.com/JLDeng/ST-Norm

**DMSTGCN**: https://github.com/liangzhehan/DMSTGCN

**STID**: https://github.com/zezhishao/STID

**ST-SSL**: https://github.com/Echo-Ji/ST-SSL

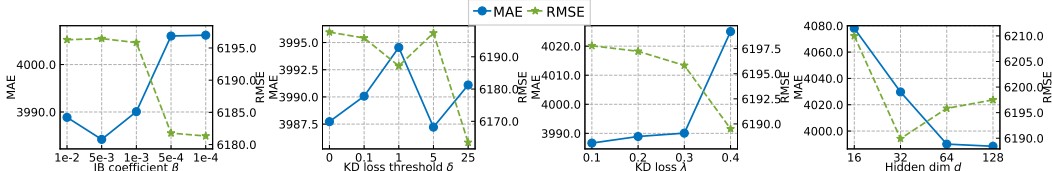

Figure 6: Hyperparameter study of our STGKD on Weather-2k.

## A.6 SUPPLEMENTARY EXPERIMENTAL RESULTS

In this subsection, we aim to address several experimental research questions. These research questions guide our exploration and evaluation of our proposed STGKD and its performance in comparison to other baselines. To investigate the impact of hyperparameters on the performance of our STGKD, we conduct a series of hyperparameter experiments. To explore the interpretability of our STGKD, we provide case studies that delve into the inner workings of the model.

### A.6.1 RESEARCH QUESTIONS ADDRESSED IN EXPERIMENTS

In the appendix, we present the experimental research questions of our study, numbered according to the suffix numbering used in the experimental section titles.

**Q1**: How does the proposed STGKD framework perform compare to state-of-the-art baselines on different experimental datasets?

**Q2**: To what extent do the various sub-modules of the proposed STGKD framework contribute to the overall performance?

**Q3**: How scalable is our STGKD for large-scale spatio-temporal prediction?

**Q4**: What is the generalization and robustness performance of our STGKD?

**Q5**: How do various hyperparameter settings influence STGKD's performance?

**Q6**: How does STGKD perform with different teacher STGNNs?

**Q7**: How is the model interpretation ability of our STGKD?

### A.6.2 VISUALIZATION OF PREDICTION

We compare the predictions of our STGKD with those of DMSTGCN and MTGNN, as well as the ground-truth values, using the PEMS traffic data. The results are visualized in Figure 7, where each figure represents a time span of one day and consists of 288 time steps.

### A.6.3 HYPERPARAMETER INVESTIGATION (Q5)

To analyze the impact of different hyperparameter configurations, we perform additional experiments where we modify a specific hyperparameter while keeping the others at their default values. We focus on four critical hyperparameters and present our experimental findings and observations based on the results obtained from the Weather dataset. The results are illustrated in Figure 6.

(i) We conduct a search for the coefficient $\beta = \beta_1 = \beta_2$ in the proposed IB principle, as defined in Equation 11. The search is performed within the range of $1e-2, 5e-3, 1e-3, 5e-4, 1e-4$. We observed that as the coefficient decreases, the Mean Absolute Error (MAE) and Root Mean Square Error (RMSE) exhibit opposite trends. After analyzing the results, we found that the best performance is achieved when $\beta = \beta_1 = \beta_2 = 1e-3$, which corresponds to the midpoint position of the coefficient range.

(ii) We explore the impact of varying the threshold $\delta$ in the bounded Knowledge Distillation (KD) loss, as defined in Equation 10. The threshold is varied within the range $0, 0.1, 1, 5, 25$. Evaluating the performance using both the Mean Absolute Error (MAE) and Root Mean Square Error (RMSE) metrics, we find that the optimal performance is achieved when $\delta = 25$.

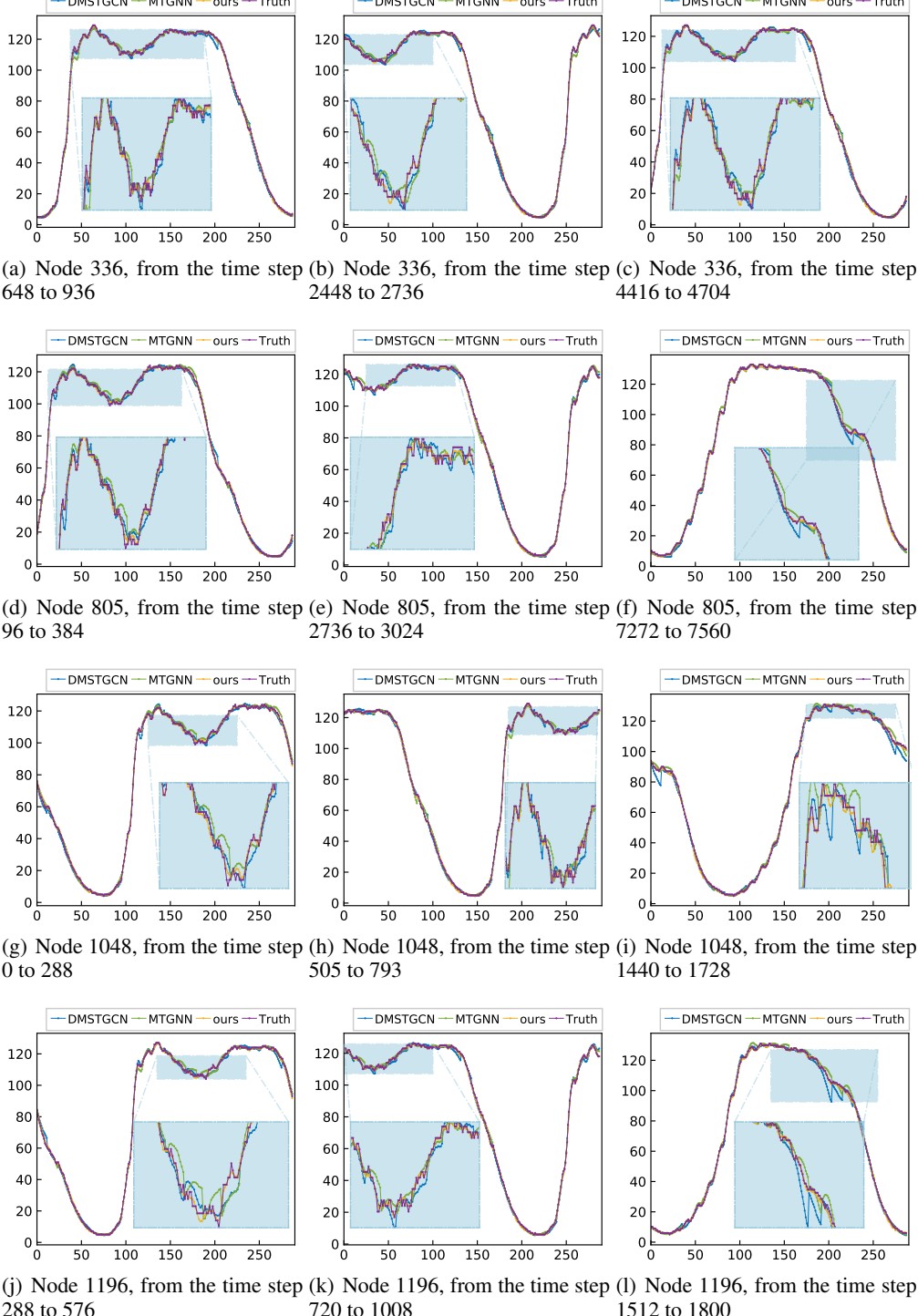

(a) Node 336, from the time step 648 to 936

(b) Node 336, from the time step 2448 to 2736

(c) Node 336, from the time step 4416 to 4704

(d) Node 805, from the time step 96 to 384

(e) Node 805, from the time step 2736 to 3024

(f) Node 805, from the time step 7272 to 7560

(g) Node 1048, from the time step 0 to 288

(h) Node 1048, from the time step 505 to 793

(i) Node 1048, from the time step 1440 to 1728

(j) Node 1196, from the time step 288 to 576

(k) Node 1196, from the time step 720 to 1008

(l) Node 1196, from the time step 1512 to 1800

Figure 7: Predictive visualization of our STGKD with other baselines on PEMS traffic data.

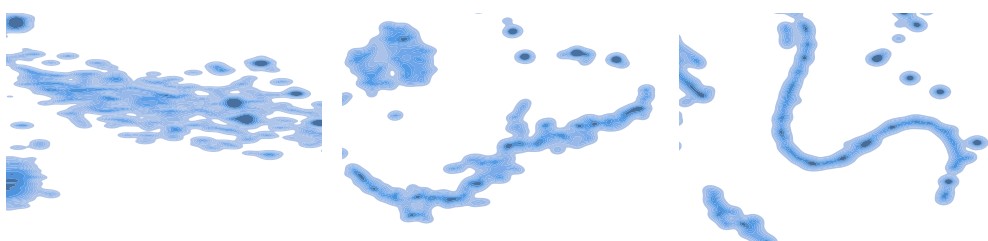

(a) Distribution of embeddings learned by our STGKD

(b) Distribution of embeddings learned by the DMSTGCN

(c) Distribution of embeddings learned by the StemGNN

Figure 8: In model interpretation evaluation, KDE visualization for distribution of embeddings learned by DMSTGCN, StemGNN and the proposed STGKD.

(iii) We investigate the influence of the coefficient $\lambda$ in controlling the loss term defined in Equation 11. The range of values for our experimental search is set to $0.1, 0.2, 0.3, 0.4$. Generally, the Mean Absolute Error (MAE) and Root Mean Square Error (RMSE) exhibit opposite trends as the coefficient varies. After analyzing the results, we find that the optimal performance is achieved when the coefficient is set to its midpoint, $\lambda = 0.3$.

(iv) We conduct a search for the dimension $d$ of hidden representations in the student MLP, with a range of $16, 32, 64, 128$. After evaluating the model's performance across these different dimensions, we find that the model performs best when the dimension $d$ is set to 64.

### A.6.4 MODEL INTERPRETATION EVALUATION WITH CASE STUDY (Q7)

To provide further insights into the learned intermediate embeddings of our STGKD and other comparative models, namely DMSTGCN and StemGNN, we visualize these embeddings in Figure 8. The visualization process involves compressing the learned embeddings into a 2-dimensional space using t-SNE dimension reduction. Subsequently, a scatter plot is generated and smoothed using Gaussian kernel density estimation (KDE) to estimate the distribution of the embeddings. Figure 8 (a) illustrates the results of our STGKD which effectively allocates different spatial regions or nodes into larger and more distinct sub-spaces. On the other hand, the baseline methods heavily rely on iterative graph information propagation, which leads to over-smoothing of node embeddings and makes them more similar. Upon examining the visualizations of the baseline methods, we observe that the STGNNs tend to over-smooth the spatial region embeddings to a significant extent, resulting in the division of regions into multiple disconnected subspaces that lack cohesion.