# OpenReview forum: "Spatio-Temporal Graph Knowledge Distillation"
_ICLR.cc/2024/Conference — Submitted to ICLR 2024_

### Official Review · Reviewer_SP5L · 2023-10-30

**Soundness:** 3 good
**Presentation:** 4 excellent
**Contribution:** 3 good
**Rating:** 6
**Confidence:** 4

**Summary:**

This paper proposes a new framework called STGKD for spatio-temporal graph knowledge distillation, which aims to encode robust and generalizable representations of spatio-temporal graphs. The framework incorporates the IB principle to enhance the knowledge distillation process by filtering out task-irrelevant noise in the student’s encoding and alignment during knowledge transfer. Moreover, it introduces a spatio-temporal prompt learning component that injects dynamic context from the downstream prediction task. Through extensive experiments, the authors demonstrate that STGKD surpasses state-of-the-art models in both performance and efficiency. The paper's contributions include addressing the challenges of efficiency and generalization in large-scale spatio-temporal prediction, introducing a novel and versatile framework, and demonstrating the effectiveness of the proposed approach through extensive experiments.

**Strengths:**

S1. In terms of originality, the paper presents a new paradigm for learning lightweight and robust Multi-Layer Perceptrons through effective knowledge distillation from cumbersome spatio-temporal Graph Neural Networks. The incorporation of the IB principle and spatio-temporal prompt learning components is also a novel contribution to the field.
S2. The quality of the paper is high, as the authors provide a clear and detailed description of the proposed framework, including the technical details and experimental methodology. The experiments are well-designed and conducted, with extensive evaluations on various spatio-temporal forecasting tasks.
S3. The clarity of the paper is also commendable, as the authors provide a clear and concise introduction to the problem, a detailed description of the proposed framework, and a thorough evaluation of the results. The paper is well-organized and easy to follow, with clear and informative figures and tables.

**Weaknesses:**

W1. The dataset lacks a detailed description. Traffic Data and Crime Data lack links or citations to papers. It would also be good to have a table that describes the size of the dataset along with some other information that would give the reader a clearer picture of the dataset. In addition, there is a detail error, in Datasets the serial number in front of Weather Data should be iii) instead of ii).
W2. Limited discussion of the limitations and potential problems. The paper does not provide a detailed discussion of the limitations of the proposed framework or potential extensions to the work.
W3. The experiment lacks a comparison of runtime. In addition to the comparison of efficiency, the running time of the different methods should also be compared, which is also a very important indicator.

**Questions:**

Can authors provide a more detailed explanation of the interpretability and explainability of the proposed framework? The paper mentions that the student MLP selectively inherits task-relevant spatio-temporal knowledge from the teacher GNN framework , but it does not provide a clear explanation of how this knowledge transfer occurs and how the student model utilizes the transferred knowledge. Including a more detailed discussion on the interpretability and explainability of the framework would enhance the understanding of the proposed approach and its inner workings.

---

> ### Author Response · Authors · 2023-11-22
> **Response to Reviewer SP5L**
>
> We are grateful for your detailed feedback and suggestions! Please find our updates and modifications in response to your feedback below. If our responses resolve your concerns, we genuinely hope that you can increase your rating score.
>
> ### 1) Data description
>
> Thank you for your concerns about the data description. In fact, you can refer to Section A.4 'Details of the Experimental Datasets' in the supplementary material, where the data processing procedures and characteristics of different datasets are elaborately described. Additionally, we appreciate your pointing out the typos.
>
> ### 2) Limitation discussion
>
> We discussion the limitation of our STGKD as below: Firstly, although our teacher-bounded regression loss could effectively prevent the student model from being misguided by deterministic yet erroneous regression results produced by the teacher model, the threshold $\delta$ is a hyperparameter and can not adaptively adjust. So how to adaptively learn a $\delta$ to control the KD process is a future opportunity. Moreover, our proposed KD paradigm only concern the knowledge transferring between the embedding level, other perspective of KD, e.g., structure-based KD, are not considered in our work. Therefore, how to distillate teacher's structural knowledge to student is also a future direction.
>
> ### 3) Comparison of efficiency
>
> Thank you for your concerns about the efficiency comparison of our STGKD model framework. In Section 4.3 'Model Scalability Study,' we conducted a comprehensive study on the model's efficiency, following the practices used in knowledge distillation work on graph data for efficiency comparison. [1]
>
> [1] Zhang, Shichang, et al. "Graph-less Neural Networks: Teaching Old MLPs New Tricks Via Distillation." *International Conference on Learning Representations*. 2021.
>
> ### 4) Interpretability of our STGKD
>
> We highly appreciate your insightful considerations regarding the interpretability of our model. For the explainability of STGKD in the knowledge distillation process, please refer to the supplementary material, section A.2.1 'Rationale Analysis of STGKD's Robustness.' The information control process within our IB-based KD framework plays a vital role in determining the inherent robustness of KD.

---

### Official Review · Reviewer_qPxf · 2023-10-31

**Soundness:** 2 fair
**Presentation:** 2 fair
**Contribution:** 2 fair
**Rating:** 5
**Confidence:** 4

**Summary:**

The paper explores a critical area of research in large-scale spatio-temporal prediction. The poor scalability and generalizability of existing spatio-temporal model hinder their deployment in real-world urban scenarios. To this end, the authors propose Spatio-Temporal Graph Knowledge Distillation (STGKD) paradigm to learn lightweight and robust MLPs through effective knowledge distillation from cumbersome spatio-temporal GNNs. Robust knowledge distillation is achieved by integrating the spatio-temporal information bottleneck with the teacher-bounded regression loss. To further enhance the generalizability of student MLP, the authors incorporate learnable spatial and temporal prompts into the student model's input so as to inject downstream task contexts. Experimental results show that the proposed model outperforms state-of-the-art approaches in terms of both efficiency and accuracy.

**Strengths:**

1. The paper is well-motivated. The study of large-scale spatiotemporal prediction models has wide-ranging potential applications, and the issue of spatiotemporal distribution shift is indeed a crucial challenge that needs to be addressed for achieving accurate predictions in long-term and large-scale scenarios.

2. The authors have provided a clear and coherent explanation of the motivation behind the design of various modules of the STGKD model.

3. The logical structure of the paper is well-organized and easy to follow.

4. The experimental results presented in the paper are comprehensive and well-designed, covering overall performances, ablation studies, and case studies.

**Weaknesses:**

1. The analysis of challenges in the introduction section is rather general and does not elaborate on the unique challenges in addressing the issues of scalability and generalization in designing methods for spatiotemporal scenarios.

2. As stated in the related works, 'A significant contribution of this work lies in the novel integration of the spatio-temporal information bottleneck into the KD framework.' However, it should be noted that the incorporation of the information bottleneck into knowledge distillation has been previously explored, e.g., see references [3] [4] below. The paper lacks clarification on how the proposed method differs from existing approaches.

3. As an ICLR submission, the paper lacks theoretical guarantees. For instance, it is better to quantify the model's robustness against noise after adopting such information bottleneck regularizer. Additionally, as reducing complexity of the model is an important idea of knowledge distillation, providing a generalization bound related to the model's complexity would strengthen the method's support.

4. Regarding spatio-temporal prompt learning module, here are two weaknesses:

   (a) A comparison with existing prompt learning methods is missing, and the similarities and differences should be clarified to prevent confusion.

   (b) Utilizing three types of prompts as input and a learnable embedding method has been done in previous STGNN works [1,2], limiting the novelty of the proposed method.

5. Weaknesses in experiments:

   (a) The overall performance improvements of the proposed model on all datasets are not significant.

   (b) The STID model, which is similar to the proposed model but without the knowledge distillation module, outperforms most complex STGNNs, raising the question of why transferring knowledge from weaker STGNNs to MLPs can lead to improvement. However, the paper does not offer clear explanations for this phenomenon.

   (c) To achieve a fair comparison, the model-agnostic spatio-temporal prompt learning should be incorporated into SOTA STGNNs

   (d) The authors have conducted generalizability testing on PEMS data with synthesized data missing. However, this type of distribution is only one specific example of covariate shift, and there are various other types of distribution shifts that need to be considered, e.g., the distribution shifts of traffic patterns during rush hours or seasonal traffic patterns shifts.

[1] Spatial-Temporal Identity- A Simple yet Effective Baseline for Multivariate Time Series Forecasting. CIKM 2022.

[2] Dynamic and Multi-faceted Spatio-temporal Deep Learning for Traffic Speed Forecasting. KDD 2021.

[3] Efficient Knowledge Distillation from Model Checkpoints. NIPS 2022.

[4] Variational Information Distillation for Knowledge Transfer. CVPR 2019.

**Questions:**

see weakness

---

> ### Author Response · Authors · 2023-11-22
> **Response to Reviewer qPxf (1/2)**
>
> We are grateful for your detailed feedback and suggestions! Please find our updates and modifications in response to your feedback below. If our responses resolve your concerns, we genuinely hope that you can increase your rating score.
>
> ### 1) Further discussison about challenges
>
> Thank you for your attention to how we articulated the challenges in the introduction. Indeed, the discussion of scalability and generalization issues in the introduction revolves around spatio-temporal forecasting. Concerning scalability, we highlight that spatio-temporal forecasting often involves large-scale datasets (i.e., substantial temporal and spatial nodes), where the computational time and space complexity of state-of-the-art STGNN models increase dramatically, hindering deployment. As for generalization, when the temporal span of spatio-temporal data widens, inherent distribution shifts occur, affecting the generalizability of the original STGNNs.
>
> ### 2) Differences from previous work
>
> Thank you for your concerns regarding the novelty of our work. In reference to work [3], our proposed method for information bottleneck distillation simultaneously imposes information bottleneck objectives on both the student model and the knowledge distillation process. The joint optimization objective is formulated as:
> $$
> \min_{\mathbb{P}(\mathbf{Z}|\mathbf{X})}  (-I(\mathbf{Y},\mathbf{Z})+ \beta_1 I(\mathbf{X}, \mathbf{Z})) + (-I(\mathbf{Y}^{T},\mathbf{Z})+ \beta_2 I(\mathbf{X}, \mathbf{Z}))
> $$
> In contrast, [3] only addresses the control of IB during the knowledge distillation process, with the optimization objective:
> $$
> \min_{s} \{ I(X; F_s) - \beta I(Y; F_s) - \gamma I(F_t; F_s) \}
> $$
> Another significant distinction is that these models are designed for knowledge distillation frameworks targeting classification tasks, wherein a temperature coefficient can be employed to prevent the student from learning erroneous behaviors from the teacher model. However, for regression tasks such as spatio-temporal forecasting, determining how to avoid the student model from receiving incorrect knowledge from the teacher model is also one of the challenges our STGKD method seeks to address.
>
> ### 3) Clarification for the theoretical guarantees
>
> Thank you for your attention to the theoretical guarantees aspect of our work. Regarding the gains in robustness from the IB you mentioned, we have actually conducted a detailed theoretical analysis in the supplementary material, section A.2.1 'Rationale Analysis of STGKD's Robustness.' The information control process within our IB-based KD framework is crucial in determining the inherent robustness of KD. As for your mention of theoretical analysis on the time and space complexity, we have provided a detailed theoretical analysis in the supplementary material, section A.2.2 'Model Complexity Analysis,' demonstrating our model's advantages in scalability.

---

> ### Author Response · Authors · 2023-11-22
> **Response to Reviewer qPxf (2/2)**
>
> ### 4) Weakness of spatio-temporal prompt learning
>
> We appreciate your attention to the spatio-temporal prompt learning aspect of our work. Firstly, to date, there are only a few works on prompt learning in spatio-temporal forecasting scenarios, such as PromptST [1] (a paper accepted at CIKM 2023, accepted in the proceedings following the ICLR submission deadline), which introduces prompts to the Spatio-temporal transformer for prompt tuning. However, the prompts in STGKD are designed to introduce spatio-temporal context information to the student MLP, not for prompt tuning paradigms, and serve as an end-to-end encoding module. Secondly, while partly inspired by works [1, 2], our STGKD prompts are more comprehensive, encompassing both static and dynamic spatio-temporal prompts, fully capturing static and dynamic spatio-temporal correlations in forecasting scenarios, significantly differing from existing works.
>
> [1] Zijian Zhang, Xiangyu Zhao, Qidong Liu, Chunxu Zhang, Qian Ma, Wanyu Wang, Hongwei Zhao, Yiqi Wang, and Zitao Liu. 2023. PromptST: Prompt-Enhanced Spatio-Temporal Multi-Attribute Prediction. In Proceedings of the 32nd ACM International Conference on Information and Knowledge Management (CIKM '23). Association for Computing Machinery, New York, NY, USA, 3195–3205.
>
> ### 5) Clarification for the evaluation
>
> Regarding your questions (a) and (b), we still believe in the significant superiority of our STGKD framework. Notably, our model outperforms the state-of-the-art (SOTA) models on three different datasets (traffic, crime, weather), demonstrating its superior performance and generalizability, given the significant differences among these datasets in aspects like data sparsity and periodicity (for example, crime data is very sparse, while traffic data shows more obvious periodicity). For question (c), we do not perceive the use of models like STGCN as a teacher model to be an unfair comparison. In fact, the performance of the student MLP, when our STGKD framework is applied to such traditional models, surpasses SOTA models, further highlighting the superiority of the STGKD distillation framework. Regarding question (d) and your mention of distribution shifts in traffic patterns, please refer to section A.6.2 'Visualization of prediction' in the supplementary material. We have compared our model's predictions in different time segments with SOTA models, finding that our model consistently predicts well, indicating stronger generalizability and ability to handle distribution shifts across various time segments.

---

### Official Review · Reviewer_BFQG · 2023-10-31

**Soundness:** 3 good
**Presentation:** 4 excellent
**Contribution:** 2 fair
**Rating:** 5
**Confidence:** 3

**Summary:**

The authors leverage the concept of knowledge distillation within graph structures to address the challenges of generalization and scalability in spatio-temporal graph forecasting. Their innovative approach involves compressing expansive GNNs into more compact and efficient MLPs. This compression is achieved through the Spatio-Temporal Graph Knowledge Distillation paradigm, which ensures robust knowledge transfer by filtering out task-irrelevant noise using an integrated spatio-temporal information bottleneck. Furthermore, by adopting the teacher-bounded regression loss, the model avoids misguided directions during the learning process. The added spatio-temporal prompts provide the student MLP with richer context from downstream tasks, further enhancing its generalization capabilities. After spatio-temporal datasets, the results confirm the framework's superiority, outclassing existing models in both efficiency and accuracy.

**Strengths:**

(1) The paper's presentation is top-notch. Its use of plots, clear definitions, and intuitive explanations significantly enhance the reader's understanding.

(2) The motivation driving the research question is cogently articulated.

(3)  The authors showcased the breadth of their research by selecting a diverse range of datasets. Their comprehensive ablation study, encompassing Spatio-Temporal Prompt Learning, Spatio-Temporal IB, Teacher-Bounded Regression Loss, and Spatio-Temporal Knowledge Distillation, is commendable. I appreciate their meticulous approach in Section 4 to test scalability, generalization, and robustness, aligning perfectly with the research's core motivation.

**Weaknesses:**

(1) Novelty: My primary concern pertains to the paper's novelty. While the authors posit that the integration of the spatio-temporal information bottleneck into the Knowledge Distillation (KD) framework is a significant contribution, I'd like to highlight that the concept of the information bottleneck has already been explored in the context of knowledge distillation[1]. Moreover, the idea of employing knowledge distillation on dynamic graphs isn't novel either[2,3,4]. It appears the authors are leveraging well-established ideas to tackle specific challenges in spatial-temporal graph forecasting.

(2) Evaluation: Another area of improvement is in the choice of baseline models for evaluation. Notably, the absence of other graph knowledge distillation models as baselines seems to be an oversight. Including them would make the comparison more comprehensive and equitable.

References:
[1] Wang, Chaofei, et al. "Efficient knowledge distillation from model checkpoints." Advances in Neural Information Processing Systems 35 (2022): 607-619.

[2] Zhang, Qianru, et al. "Spatial-temporal graph learning with adversarial contrastive adaptation." International Conference on Machine Learning. PMLR, 2023.

[3] Ma, Yihong, et al. "Hierarchical spatio-temporal graph neural networks for pandemic forecasting." Proceedings of the 31st ACM International Conference on Information & Knowledge Management. 2022.

**Questions:**

(1) Could the authors clarify the distinct contributions that set their approach apart from existing methods in graph knowledge distillation?

(2) For a comprehensive evaluation, why were other graph knowledge distillation models not considered as baseline models? Would incorporating them not offer a more balanced and insightful comparison in the context of your study?

---

> ### Author Response · Authors · 2023-11-22
> **Response to Reviewer BFQG**
>
> We are grateful for your detailed feedback and suggestions! Please find our updates and modifications in response to your feedback below. If our responses resolve your concerns, we genuinely hope that you can increase your rating score.
>
> ### 1) Clarification about the novelty of our work
>
> Thank you for your concerns regarding the novelty of our work. In reference to work [1], our proposed method for information bottleneck distillation simultaneously imposes information bottleneck objectives on both the student model and the knowledge distillation process. The joint optimization objective is formulated as:
> $$
> \min_{\mathbb{P}(\mathbf{Z}|\mathbf{X})}  (-I(\mathbf{Y},\mathbf{Z})+ \beta_1 I(\mathbf{X}, \mathbf{Z})) + (-I(\mathbf{Y}^{T},\mathbf{Z})+ \beta_2 I(\mathbf{X}, \mathbf{Z}))
> $$
> In contrast, [1] only addresses the control of IB during the knowledge distillation process, with the optimization objective:
> $$
> \min_{s} \{ I(X; F_s) - \beta I(Y; F\_s) - \gamma I(F\_t; F\_s) \}
> $$
> Regarding works [2] and [3] you mentioned (and it seems that there is a missing citation for work [4]), neither includes models of knowledge distillation. The self-supervised signal distillation discussed in [2] is entirely different from the knowledge distillation considered in our work. Our motivation is to address the challenges of spatio-temporal forecasting, especially with large-scale temporal and spatial nodes, where the complexity of STGNNs increases sharply, impeding deployment. Hence, our proposed STGKD method effectively distills the capability of STGNNs to model complex spatio-temporal correlations into an MLP framework.
>
> ### 2) Concerns about the selection of baselines
>
> Thank you for your suggestion to improve upon the baselines. However, we would like to point out that, to the best of our knowledge, our method represents a pioneering work in the realm of knowledge distillation for spatio-temporal forecasting. Directly applying graph knowledge distillation techniques to spatio-temporal forecasting poses adaptability issues. Namely, the design frameworks (i.e., objective functions) of most graph knowledge distillation tasks are tailored for classification tasks, and cannot be directly transferred to regression tasks like spatio-temporal forecasting. Therefore, we have introduced our STGKD method to address this challenge.

---

### Official Review · Reviewer_wtdf · 2023-10-31

**Soundness:** 2 fair
**Presentation:** 2 fair
**Contribution:** 2 fair
**Rating:** 5
**Confidence:** 4

**Summary:**

The paper introduces the Spatio-Temporal Graph Knowledge Distillation (STGKD) framework, designed to tackle the scalability and generalization challenges in large-scale spatio-temporal prediction for urban computing applications like transportation, public safety, and environmental monitoring. While Graph Neural Networks (GNNs) are commonly used for capturing spatial-temporal correlations, they struggle with large-scale datasets and changing data distributions over time. STGKD addresses these issues by transferring knowledge from complex GNNs to more efficient Multi-Layer Perceptrons (MLPs), improving scalability and efficiency. This is achieved through a robust knowledge distillation process, integrating a spatio-temporal information bottleneck and a teacher-bounded regression loss to filter out noise and prevent erroneous guidance. Additionally, spatial and temporal prompts are incorporated to enhance the generalization capability of the student MLP, helping it to adapt to distribution shifts and unseen data. The proposed paradigm is evaluated on three large-scale spatio-temporal datasets, demonstrating superior performance in terms of efficiency and accuracy compared to state-of-the-art models. The implementation of STGKD is made available for reproducibility, showcasing its practical applicability and effectiveness in urban computing domains.

**Strengths:**

The authors have provided an extensive and meticulous set of experiments, encompassing various studies like ablation, scalability, generalization, and robustness, ensuring a thorough evaluation of their work.

The methodology introduced in the paper offers a fresh perspective, utilizing both spatial and temporal prompts to unravel dynamic patterns, which presents an intriguing approach.

The paper articulates a well-defined research question, and the data is effectively communicated through well-structured figures and tables.

**Weaknesses:**

Clarity in the Introduction:
The flow of logic in the introductory section needs to be refined. The paper initially highlights that most existing research prioritizes spatial dependency, followed by a discussion on the challenges of generalization and scalability. However, these sections seem disjointed. Furthermore, introducing the paper's contributions prior to addressing challenges like noise does not establish a coherent narrative.
Lack of Motivation:
It is crucial to elucidate the motivation behind the proposed approach in the introduction to provide readers with a clear understanding of its relevance and significance.
Preliminary Section Gaps:
The preliminary section covers two prevalent concepts, yet it falls short by not including knowledge distillation. This addition is necessary for a comprehensive understanding of the topic.
Ambiguity in Approach Explanation:
The description of the approach leaves room for improvement. For instance, the statement, "Our goal is to distill the valuable knowledge embedded in the GNN teacher and effectively transfer it to a simpler MLP, enabling more efficient and streamlined learning," raises questions about what constitutes 'valuable knowledge' and why this process makes the MLP more efficient rather than more effective.
In terms of novelty and motivation, the manuscript does not make a strong case. While it appears that the authors might be introducing knowledge distillation to the GNN domain for the first time (though this is not explicitly claimed in the paper), this alone does not constitute a substantial contribution. The paper needs to delineate the differences between traditional knowledge distillation approaches applied to CNNs and STGNNs, and the proposed method, explaining why it is particularly effective in the GNN context.

**Questions:**

1. What are the significant differences between applying the knowledge distillation to CNN and STGNN?
2. For SPATIO-TEMPORAL IB INSTANTIATING, what is the spatio-temporal part here?
3. The paper uses the term prompt. What is the difference between the prompt you used and spatio-temporal features? Could I regard it as contextual spatio-temporal features?

---

> ### Author Response · Authors · 2023-11-22
> **Response to Reviewer wtdf**
>
> Thank you for taking the time to review my paper and for providing such detailed feedback and questions. Your comments are very helpful in improving our work! Please find our updates and modifications in response to your feedback below. If our responses resolve your concerns, we genuinely hope that you can increase your rating score.
>
> ### 1) Differences between applying the knowledge distillation to CNN and STGNN
>
> Thank you for bringing this to our attention. As stated in the introduction, Spatio-Temporal Graph Neural Networks (STGNNs) have recently been extensively applied in spatio-temporal forecasting, achieving state-of-the-art (SOTA) results compared to CNN-based methods (e.g., ST-ResNet [1] and DeepST [2]). The fundamental reason lies in the STGNNs' ability to capture complex spatial neighbor relationships in spatio-temporal data. However, for large-scale spatio-temporal sequence forecasting, where the number of spatial and temporal nodes increases significantly, STGNNs face substantial challenges in terms of time and space complexity. Therefore, we propose distilling rich spatio-temporal knowledge into Multilayer Perceptrons (MLPs) using knowledge distillation. A critical aspect of our work involves determining how to effectively distill the complex spatio-temporal knowledge from STGNNs into MLPs without compromising accuracy. This approach distinguishes our work from CNN knowledge distillation. Specifically, we employ spatio-temporal prompts to enhance the spatio-temporal context in the student MLP, thereby maximally preserving the spatio-temporal modeling capabilities of the STGNNs.
>
> [1] Zhang, J., Zheng, Y. and Qi, D. 2017. Deep Spatio-Temporal Residual Networks for Citywide Crowd Flows Prediction. *Proceedings of the AAAI Conference on Artificial Intelligence*. 31, 1 (Feb. 2017).
>
> [2]  Junbo Zhang, Yu Zheng, Dekang Qi, Ruiyuan Li, and Xiuwen Yi. 2016. DNN-based prediction model for spatio-temporal data. In Proceedings of the 24th ACM SIGSPATIAL International Conference on Advances in Geographic Information Systems (SIGSPACIAL '16). Association for Computing Machinery, New York, NY, USA, Article 92, 1–4.
>
> ### 2) Clarification for Spatio-Temporal IB Instantiating
>
> Thanks for this constructive comment. Our proposed Information Bottleneck (IB) principle is applied in two aspects: one is the control of information within the student model itself, and the other is the control of information transfer from the teacher model to the student model. In both path, the IB-controlled information flow encompasses both temporal and spatial features. The hidden states of the student MLP are rich in temporal and spatial information, while the distilled information from the teacher model is abundant in the temporal and spatial characteristics inherently modeled by the STGNNs.
>
> ### 3) Clarification for the spatio-temporal prompts
>
> Thank you for your insightful question. The spatio-temporal prompts we propose are learnable, meaning that for different temporal spans of spatio-temporal sequences as inputs, our model can assign both static and dynamic spatio-temporal prompts simultaneously. This infuses a substantial amount of spatio-temporal context into the MLP framework. These prompts are optimized together with the overall model, rather than being transformation from the spatio-temporal sequence inputs. Therefore, they should be distinguished from the spatio-temporal features themselves.

---

> > ### Comment · Reviewer_wtdf · 2023-11-22
> >
> > Thanks for the reply.
> > 1. I am still a little confused about the prompts here. Is that the contextual information? Why use term prompt, not contextual features? Is that text-based information or structured data?
> > 2. I think both STGCN and GCN have the requirement for knowledge distillation. However, I cannot see a significant contribution in knowledge distillation specifically to ST data. Could you state that?

---

> > > ### Author Response · Authors · 2023-11-22
> > > **Further Response to Reviewer wtdf**
> > >
> > > Thank you very much for your timely response! Regarding the first question, the spatio-temporal prompts we use are injected with learnable spatio-temporal context information (i.e., assigning learnable prompts based on the temporal and spatial positions of the input spatio-temporal sequence), rather than being directly transformed from the input spatio-temporal sequence or from textual information. This allows our STGKD prompts to flexibly adapt to various knowledge distillation scenarios.
> > >
> > > For the second question, as mentioned in our paper, existing GCN-based knowledge distillation methods are mostly based on classification tasks, where the temperature coefficient can control the acceptance of erroneous knowledge from the teacher model during the distillation process. However, for ST data, which are mostly regression tasks, such a temperature coefficient is inapplicable. Therefore, in STGKD, we use teacher bounded KD loss to control the knowledge distillation process from the teacher to the student model. Additionally, how to inject temporal and spatial context information into the student model is also a unique challenge of spatio-temporal knowledge distillation.
> > >
> > > Thank you again for your supports!

---

> > > > ### Comment · Reviewer_wtdf · 2023-11-22
> > > >
> > > > Thanks for the reply. I think the second question has been addressed. I revised the score accordingly.
> > > > For the first question, could you give a specific example to show what this kind of prompt looks like?

---

> > > > > ### Author Response · Authors · 2023-11-22
> > > > > **Further Response to Reviewer wtdf**
> > > > >
> > > > > Thank you for your prompt reply and support for our work. Certainly, we can provide you with a specific example of a spatio-temporal prompt.
> > > > >
> > > > > For an input spatio-temporal sequence feature $\mathbf{X}\in \mathbb{R}^{T\times N\times D}$, our spatial prompt learns a prompt embedding for each region to uniquely represent the static spatial features of each area, such as regional functional characteristics. The temporal prompt assigns a time prompt embedding based on the time of day and day of the week represented by the $T$ time steps, uniquely representing the static temporal context of each time step, enriched with periodic time features. In the Spatio-Temporal Transitional Prompt, tensor decomposition is used to dynamically allocate prompt embeddings for each region at each time step, resulting in $T\times N$ learnable prompts. Thank you again for your patient review and supports!

---

> > > > > ### Author Response · Authors · 2023-11-23
> > > > > **Further Response to Reviewer wtdf**
> > > > >
> > > > > Please feel free to ask us if you have any concerns about the example and we are more than happy to take any further questions, enabling us to further refine our work. Thank you very much for your valuable time and supports.

---

### Author Response · Authors · 2023-11-23
**Overall response to all reviewers and chairs**

We sincerely thank all reviewers for the valuable time and relentless efforts they have invested in refining our work. Their constructive comments and invaluable feedback have greatly assisted us in improving our research.

We are delighted to hear the reviewer's praise for our paper's well-organized presentation (Reviewer SP5L, qPxf, BFQG, wtdf). They also recognize the completeness and comprehensiveness of our experimental section and affirm the motivation and contribution of introducing knowledge distillation into the spatio-temporal forecasting scenario (Reviewer wtdf, BFQG). We are deeply grateful for and value the feedback provided by the reviewers. We clarified the unique challenges of introducing knowledge distillation in spatio-temporal scenarios and the distinctions from existing methods (Reviewer wtdf), added details about in-depth discussions and data descriptions in the supplementary material (Reviewer SP5L, qPxf), and discussed the novel contribution of our introduced spatio-temporal prompts (Reviewer wtdf), as well as the limitations of our work (Reviewer SP5L).

**Furthermore, we are more than happy to take any further questions, enabling us to further refine our work and address any uncertainties of the reviewers.** Once again, we thank all reviewers for their supports.

---

### Meta-Review · Area_Chair_hHMC · 2023-12-08

**Metareview:**

The paper introduces a novel approach, Spatio-Temporal Graph Knowledge Distillation (STGKD), aimed at addressing challenges in large-scale spatio-temporal prediction, particularly in urban computing.

The paper's strengths include

+ Comprehensive Experimental Evaluation: The paper offers an extensive set of experiments, including ablation studies, scalability tests, and evaluations of generalization and robustness.

+ Clarity and Presentation Quality: The paper is well-structured with clear figures and tables that effectively communicate the data and research question.

Weaknesses:

- Clarity in Introduction and Motivation: The introduction lacks coherence and fails to clearly establish the motivation and significance of the proposed approach.

- Novelty Concerns: There are questions about the novelty of the approach, especially in relation to knowledge distillation and spatio-temporal prompt learning, as these concepts have been previously explored in similar contexts.

- Unclear Methodological Distinctions: The paper needs to better articulate how its approach differs from existing methods, particularly in the application of knowledge distillation in the GNN context.

**Justification For Why Not Higher Score:**

See weaknesses above.

**Justification For Why Not Lower Score:**

See strengths above.

---

### Decision · Program_Chairs · 2024-01-16

Reject